# Ultra-durable cell-free bioactive hydrogel with fast shape memory and on-demand drug release for cartilage regeneration

Yuxuan Yang [1,5] ✉, Xiaodan Zhao[1,5], Shuang Wang[1], Yanfeng Zhang[2], Aiming Yang[3], Yilong Cheng [2,3] ✉ & Xuesi Chen [4]

Osteoarthritis is a worldwide prevalent disease that imposes a significant socioeconomic burden on individuals and healthcare systems. Achieving cartilage regeneration in patients with osteoarthritis remains challenging clinically. In this work, we construct a multiple hydrogen-bond crosslinked hydrogel loaded with tannic acid and Kartogenin by polyaddition reaction as a cell-free scaffold for in vivo cartilage regeneration, which features ultra-durable mechanical properties and stage-dependent drug release behavior. We demonstrate that the hydrogel can withstand 28000 loading-unloading mechanical cycles and exhibits fast shape memory at body temperature (30 s) with the potential for minimally invasive surgery. We find that the hydrogel can also alleviate the inflammatory reaction and regulate oxidative stress in situ to establish a microenvironment conducive to healing. We show that the sequential release of tannic acid and Kartogenin can promote the migration of bone marrow mesenchymal stem cells into the hydrogel scaffold, followed by the induction of chondrocyte differentiation, thus leading to full-thickness cartilage regeneration in vivo. This work may provide a promising solution to address the problem of cartilage regeneration.

As a globally prevalent and disabling disease, osteoarthritis (OA) imposes a significant socioeconomic burden on both individuals and healthcare systems, and over $16.5 billion has been spent annually on OA treatment[1]. It has been reported that more than 7% of the global population suffers from OA, and the number of OA patients has been elevated by 48% in the last two decades[2,3]. The main pathogenesis of OA is progressive loss and destruction of articular cartilage, which could eventually lead to joint deformation and immobility[4]. However, the regenerative capability of native articular cartilage is limited due to the absence of blood vessels, lymphatics, and nerves as well as the harsh biomechanical environment. Therefore, the repair of articular cartilage defects remains a great challenge clinically.

Current strategies for the restoration of cartilage functions include microfracture, mosaicplasty, cartilage transplantation, and medication treatment[5–7]. Nevertheless, these treatments may cause significant tissue damage and exhibit only limited curative effects[8,9], which may lead to the employment of arthroplasty or total joint replacement for most advanced OA patients[10]. Tissue engineering utilizing stem cell-loaded scaffolds has been recognized as a promising method for cartilage regeneration. Previous studies showed that mesenchymal stem cell (MSC) loaded scaffolds could promote cell proliferation and exhibit promising therapeutic effects on cartilage regeneration[11–13]. Owing to their similar properties to the natural extracellular matrix (ECM) (high water content, porous structure,

[1]Key Laboratory of Shaanxi Province for Craniofacial Precision Medicine Research, College of Stomatology, Xi'an Jiaotong University, Xi'an 710049, China. [2]School of Chemistry, Xi'an Jiaotong University, Xi'an 710049, China. [3]Department of Nuclear Medicine, The First Affiliated Hospital of Xi'an Jiaotong University, Xi'an Jiaotong University, Xi'an 710061, China. [4]Key Laboratory of Polymer Ecomaterials, Changchun Institute of Applied Chemistry, Chinese Academy of Sciences, Changchun 13022, China. [5]These authors contributed equally: Yuxuan Yang, Xiaodan Zhao. ✉e-mail: yangyuxuan@xjtu.edu.cn; yilongcheng@mail.xjtu.edu.cn

and biocompatibility)[14], hydrogel scaffolds have been widely investigated in tissue engineering, in which injectable hydrogels with the potential for minimally invasive surgery and facile stem cells and drug loading capacities have been verified to have a positive effect on the stimulation of cartilage regeneration[15,16]. However, cell-laden hydrogels usually show low mechanical properties and cannot maintain their structural integrity by endoscopy delivery systems, which makes them unsuitable for arthroscopy treatment. Moreover, stem-cell-based treatment is still associated with multiple concerns, such as poor cell survival, immune rejection, complex cell culture procedures, decreased regeneration capacity after transplantation, and ethical concerns[17,18].

As an alternative solution to address the abovementioned issues, the introduction of cell-homing agents into scaffolds to recruit MSCs from subchondral bone for cartilage regeneration is an attractive option that could avoid the usage of autologous cartilage tissue and reduce immune rejection[17,19,20]. The current cell recruitment strategies are normally based on cytokines or functional peptides which are costly and may cause potential disturbance to subchondral bone homeostasis or lead to cartilage calcification[21]. Furthermore, the mechanical strength of the scaffolds is also essential since the matched mechanical properties with cartilage tissue could promote the chondrogenic differentiation of recruited MSCs[22,23]. Therefore, ideal cell-free hydrogel scaffolds for cartilage repair should feature the following advantages: the capability for minimally invasive surgery, adequate tissue adhesiveness for local fixation, similar mechanical strength to cartilage tissue, stem cell recruitment, and chondrocyte differentiation.

Recent studies have revealed that tannic acid (TA) could promote cell migration in different cell lines, including epithelial and mesenchymal stem cells[24–26]. TA is a plant-derived natural polyphenol with multiple bioactivities. Thus, the incorporation of TA with hydrogel matrix for cartilage engineering should not only induce targeted MSC migration but also regulate inflammation and reduce oxidative stress in situ for tissue regeneration. Kartogenin (KGN), a small hydrophobic molecule, has been verified to promote MSC differentiation into chondrocytes at the proper dose[6,27]. Therefore, based on their solubility in aqueous environments, the combination of TA and KGN in hydrogels could realize on-demand drug release to sequentially recruit MSCs and stimulate chondrocyte differentiation to accelerate new cartilage formation.

In this study, we reported an ultra-durable hydrogel with multiple bioactivities and fast shape memory properties for cell-free cartilage regeneration. A hydrogen bonding reinforced factor, imidazolidinyl urea (IU), was introduced into polyurethane to produce multiple hydrogen-bond mediated hydrogel (PMI) by polyaddition with poly (ethylene glycol) (PEG) and methylene diphenyl 4,4-diisocyanate (MDI) (Supplementary Fig. 1). TA and KGN were simultaneously loaded into PMI hydrogel network to afford the formation of the targeted hydrogel (PTK). Owing to multiple hydrogen bonds formed by the urethane group, IU, and TA in the hydrogel network, the PTK hydrogel showed a high fracture strength of 1.1 MPa, stable mechanical properties during 28000 tensile cycles, and promising shape memory capability, which made it suitable for minimally invasive surgery and maintained the network integrity during constant mechanical stimulation in joint movement. Moreover, the introduction of TA endowed the PTK hydrogel with adequate tissue adhesiveness for in situ immobilization, and the preferred release after in vivo implantation could suppress inflammatory reactions, clear overexpressed reactive oxygen species (ROS), and prevent potential bacterial colonization to reestablish an appropriate microenvironment for bone marrow mesenchymal stem cell (BMSC) homing (Fig. 1). Furthermore, with the gradual degradation of the PTK hydrogel, hydrophobic KGN was released in a sustained manner, which could promote the differentiation of recruited BMSCs into chondrocytes for cartilage regeneration.

## Results

### Synthesis and characterization of the hydrogels

As shown in Supplementary Fig. 1, the pristine polymer (PMI) was synthesized by polyaddition of MDI, IU, and PEG using dibutyltin dilaurate (DBTDL) as a catalyst. After polymerization, KGN was mixed with polymer solution followed by vacuum drying to form a drug-loaded polymeric film (PKG film). Profiting from the formation of multiple hydrogen bonds by urethane and urea groups in the polymer chains, PMI and PKG hydrogels were obtained after swelling the polymer films in water[28,29]. Meanwhile, PMI and PKG films were immersed in TA solution (5% w/v) to afford the formation of PTA and PTK hydrogels with additional hydrogen bonds between TA and PMI. The compositions of PMI, PKG, PTA, and PTK hydrogels as well as the noncovalent interactions in the hydrogel networks are shown in Supplementary Fig. 2. It was found that the color of PTA and PTK hydrogels was faint yellow due to the presence of TA, and there was no obvious difference between PMI and PKG hydrogels. The formation of multiple noncovalent interactions was then investigated by FT-IR spectroscopy as presented in Supplementary Fig. 3. Compared with the PTA and PTK hydrogels, the absorption bands of the C=O and O-H stretch in TA shifted from 1637 to 1714 cm$^{-1}$ and 3285 to 3326 cm$^{-1}$, respectively, indicating the formation of hydrogen bonds between PMI and TA[30]. However, it was difficult to detect the characteristic peaks of KGN in PTK hydrogel due to the low concentration. The scanning electron microscopy (SEM) images showed that all the hydrogels featured porous structures with connections, and the size of the pore was reduced for the PTA and PTK hydrogels compared with the PMI and PKG hydrogels (Supplementary Fig. 4), which was due to the improved crosslinking density resulting from the extra hydrogen bonds formed in the network between TA and PMI. In the same manner, the introduction of KGN did not show any effect on the hydrogel microstructure.

Previous studies showed that the mechanical properties of the implanted scaffold imposed a significant impact on chondrogenesis during cartilage matrix formation[31]. The stiffness of the substrates can alter cell response and behavior through biomechanical stimulation[32,33]. In this work, we found that the introduction of TA into PMI hydrogel could significantly improve the mechanical properties of the hydrogels. According to the tensile tests in Fig. 2a, the fracture stress of the PTA and PTK hydrogels was as high as 1.1 MPa. The presence of KGN did not show an obvious effect on the mechanical properties of the hydrogels, in which there were no significant differences between PMI and PKG hydrogels in fracture stress and strain. Although the fracture and compression stresses of the PTK hydrogel were lower than that of natural porcine cartilage (Supplementary Fig. 5), they were still higher than those hydrogels reported in previous work for cartilage regeneration[15,16]. The implanted materials should be able to withstand mechanical loading under physiological movements[23,34]. Successive loading-unloading examinations were then conducted on PMI and PTK hydrogels to investigate their fatigue-resistance property (Fig. 2b). It was also found that the introduction of TA significantly increased the antifatigue property of the PTK hydrogel in successive loading-unloading cycles (28,000 cycles), while the PMI hydrogel broke down at approximately 1600 cycles. The change in the maximum stress and the retention ratio of the initial stress at 0, 20, 200, 1000, 3000, 6000, 10,000, and 20,000 cycles by the PTK hydrogel were quantitatively analyzed (Supplementary Fig. 6 and Fig. 2c). We found that the maximum tensile stress of the PTK hydrogel was constant for 20000 cycles, and more than 90% of the original value was retained; however, the PMI hydrogel was broken after 1600 cycles, demonstrating the excellent cyclic stability and durability of the PTK hydrogel.

The adhesive property of the PTK hydrogel was quantitatively evaluated by lap shear test with cartilage tissues (Supplementary Fig. 7). The results showed that the adhesive strength of cartilage tissue

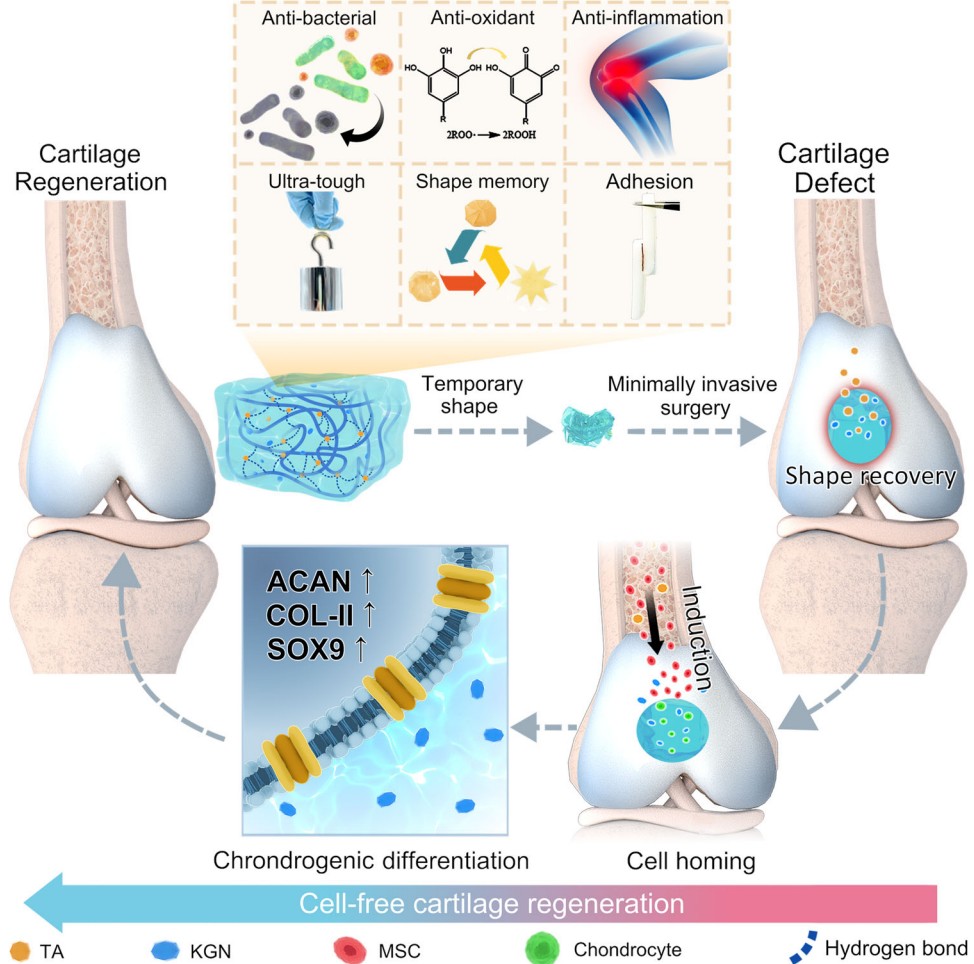

**Fig. 1 | Schematic demonstration of PTK hydrogel-induced cell-free cartilage regeneration.** PTK hydrogel with multiple hydrogen-bonds crosslinked network features ultra-durable mechanical properties, adequate adhesiveness with cartilage tissues, anti-inflammation, antioxidant, and antibacterial capabilities, and the fast shape memory property endows the hydrogel with potential for minimally invasive surgery. After implantation in the cartilage defect, the stage-dependent release of TA and KGN can sequentially recruit BMSCs migration into the hydrogel matrix and induce chondrocyte differentiation for neocartilage formation with the expression of related proteins (ACAN, COL II, and SOX9).

with the hydrogel was as high as 19.2 kPa (Fig. 2d), which can ensure the entire filling of the defect and maintain an appropriate microenvironment for cartilage regeneration[35]. Moreover, there was no obvious difference in the adhesive strength found between the PTA and PTK hydrogels, suggesting that the introduction of KGN did not affect the adhesion behavior.

## Shape-memory properties of PTK hydrogel

To develop an implanted hydrogel used in minimally invasive surgery, the materials should be fixed as a temporary shape to be transferred through a narrow tube or syringe (mimic the arthroscope instrument) for facile manipulation and can be instantly recovered to the initial shape upon in vivo stimuli to fill the defect and further promote tissue regeneration. Profiting from the dynamic hydrogen bond crosslinking network, the PTK hydrogel exhibited thermo-triggered fast shape memory properties. As shown in Fig. 2e, PTK hydrogel with a strip shape (30 mm in length) was first twisted on a glass rod and fixed at 4 °C to form the temporary helix shape, which can be transferred through a glass tube with a diameter of 8 mm. When the hydrogel was heated to 37 °C, the original shape was recovered within 30 s. The same phenomenon was also observed with the temporary coil shape (Fig. 2f). The changes in the central angle (θ) and diameter (D) of the PTK hydrogel during the shape recovery process were documented to further estimate the shape memory properties (Supplementary Fig. 8).

It was found that D and θ changed from 8 to 29 mm and 710° to 30° for the helix-shaped PTK hydrogel and 5 to 25 mm and 1064° to 53° for the coil-shaped PTK hydrogel in 30 s at 37 °C (Supplementary Fig. 9), respectively, which indicated its fast shape recovery capability. The shape recovery ratio ($R_r$) was then measured based on the reported protocol[36]. As shown in Supplementary Fig. 10, after 9 cycles, the $R_r$ of the PTK hydrogel was almost 100%. These results verified the vast potential of PTK hydrogel for minimally invasive procedures.

## In vitro drug release, in vitro and in vivo biocompatibility

In this work, UV spectroscopy and high-performance liquid chromatography (HPLC) were employed to investigate TA and KGN release profiles from PTK hydrogel. As shown in Fig. 3a, 40% of TA was released in the first 6 h and 86% of TA could be released in the first week. The burst release of TA in a short time was attributed to the following aspects: 1. Some TA is free in the hydrogel matrix and could be released first due to its highly hydrophilic nature. 2. TA was also encapsulated in the hydrogel matrix by hydrogen bonding interactions, and the presence of different salts in the release buffer (PBS) could disrupt the physical interactions and lead to the fast release of TA in the surface layer. 3. Although TA could form multiple hydrogen bonds with PMI, single or double hydrogen bonds could still be formed with relatively weak interaction, which could be easily broken and lead to free TA release. The TA inside the hydrogel matrix with multiple

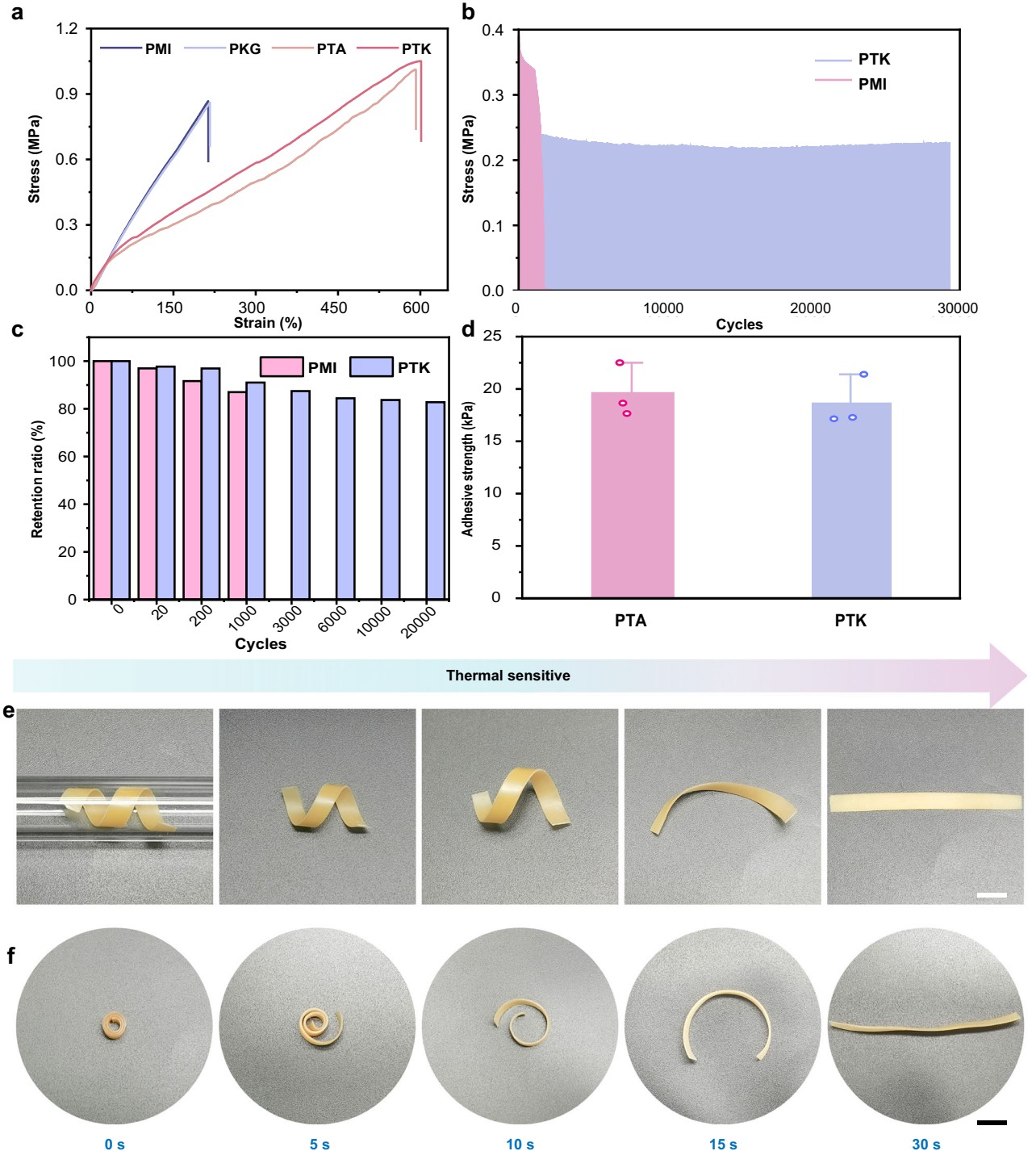

**Fig. 2 | Mechanical and shape-memory properties of PTK hydrogel. a** Tensile stress-strain curves of PMI, PKG, PTA, and PTK hydrogels. **b** Successive loading-unloading test of PMI and PTK hydrogels. **c** The retention ratio of the initial stress at the 0, 20, 200, 1000, 3000, 6000, 10000, and 20000 cycles during the successive loading-unloading test of PMI and PTK hydrogel. **d** The lap-shear adhesive strength of PTA and PTK hydrogels with cartilage. Photographs of the shape-memory performances of PTK hydrogel with the temporary helix shape (**e**) and coil shape (**f**). Scale bar: 5 mm. Data in **d** are presented as mean values ± SD. (*n* = 3 independent samples).

hydrogen bonds with PMI was further released in a sustained manner. The HPLC results showed that KGN showed continuous release for 30 days, and approximately 60% of KGN was released after 60 days.

The in vitro biocompatibility of the PTK hydrogel was evaluated via MTT assay, Live/Dead staining, and blood compatibility. As presented in Fig. 3b, the PMI, PKG, PTA, and PTK hydrogels did not show cytotoxicity to BMSCs, and more than 90% cell viability was achieved in all the experimental groups compared with the control group,

indicating the ideal cytocompatibility of PMI-based hydrogels. The Live/Dead staining results were consistent with the MTT results. After 24 h of incubation, the cells in all hydrogel-treated groups showed similar green fluorescence to the control group, and the cell number increased significantly after 96 h of cultivation, suggesting that the hydrogels did not show any side effects on cell proliferation (Fig. 3c). In the blood compatibility study, all the hydrogels presented negligible hemolysis compared to obvious hemoglobin release in the Triton-X

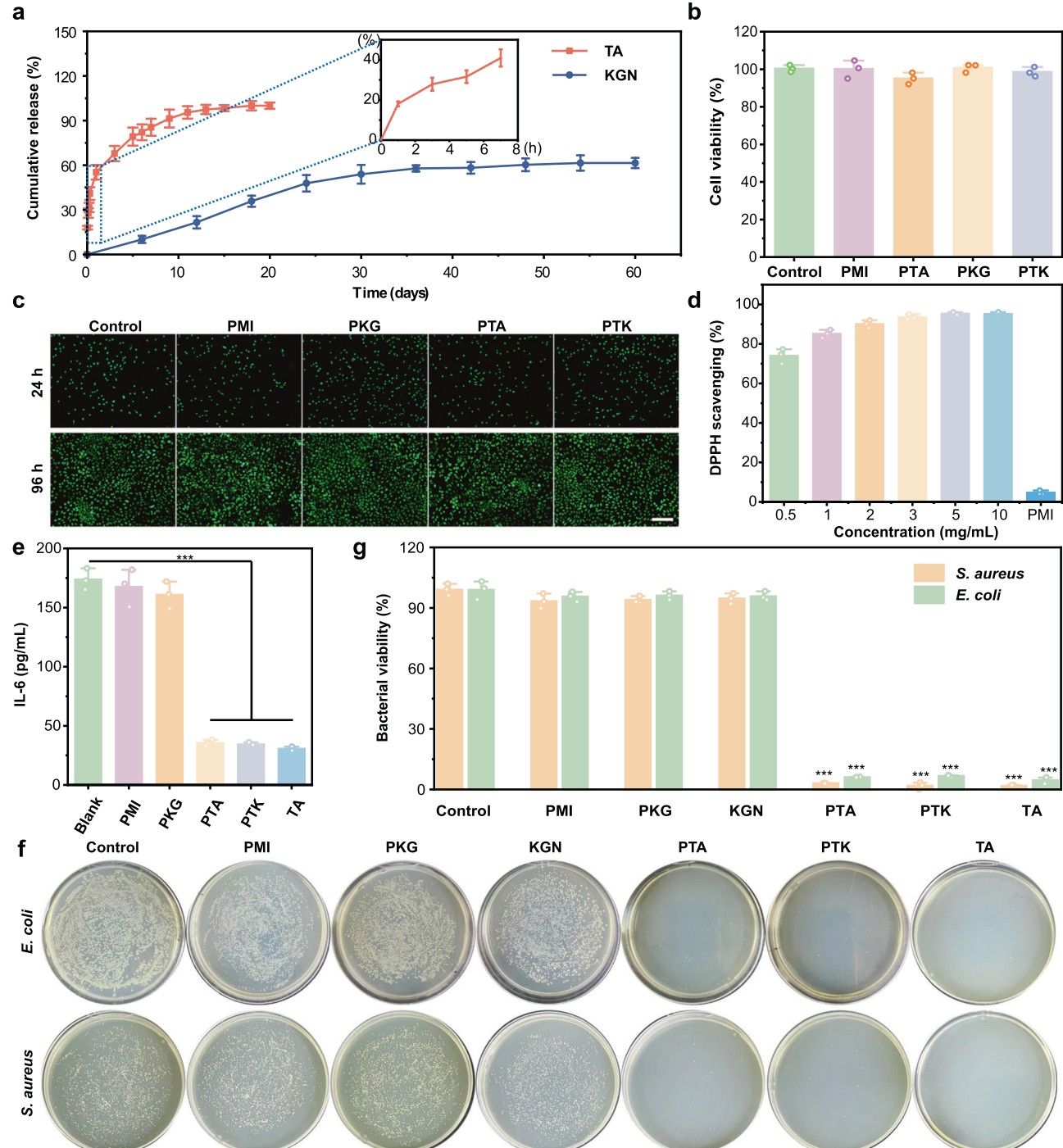

**Fig. 3 | Drug release profiles and bioactive properties of PTK hydrogel.**
**a** Cumulative TA and KGN release profiles from PTK hydrogel. **b** Cell viability treated with PMI, PTA, PKG, and PTK hydrogels. **c** Live/Dead assay of BMSCs incubated with PMI, PTA, PKG, and PTK hydrogels for 24 or 96 h. Scale bar: 200 μm. **d** Antioxidative efficiency of PTK hydrogel with different concentrations. **e** In vitro anti-inflammation of PTK hydrogel by IL-6 ELISA test. **f** Photographs of the survival of *E. coli* and *S. aureus* clones after incubation with PBS, KGN, TA, PMI, PKG, PTA, and PTK hydrogels for 12 h. Scale bar: 2 cm. **g** Quantitative antibacterial efficiency of PBS, KGN, TA, PMI, PKG, PTA, and PTK hydrogels. Values in **a**, **b**, **d**, **e**, and **g** represent mean ± SD. One-way ANOVA test. Source data are provided as a Source data file. ($n = 3$ independent samples. *$P < 0.05$, **$P < 0.01$, ***$P < 0.001$).

treated group (Supplementary Fig. 11), which showed the excellent blood compatibility of all the hydrogels. The long-term biocompatibility evaluation of the hydrogels was conducted by in vivo subcutaneous implantation to investigate the toxicity to the main organs (heart, kidney, liver, lung, and spleen) of rats. Two months after implantation, the main organs of the rats were collected and histologically evaluated. As shown in Supplementary Fig. 12, no obvious toxic

reaction was identified in all groups, indicating the good biocompatibility of the hydrogels.

As a material developed for in vivo applications, the tissue reaction is crucial for biocompatibility evaluation. The in vivo degradation and tissue biocompatibility of the hydrogels were evaluated by subcutaneous implantation in Sprague Dawley (SD) rats. The PMI and PTK hydrogels were implanted under the dorsal skin of rats. The residual

hydrogels were collected and weighed at predetermined time intervals. As depicted in Supplementary Fig. 13, the degradation rate of the PTK hydrogel was slower than that of the PMI hydrogel, which may be due to the dense network mediated by hydrogen bonds between PMI and TA. The degradation of PTK hydrogel could last for 60 days, which is coordinated with the cartilage recovery time[37,38]. After 7 and 30 days of implantation, the dorsal skin around the implantation area was collected for H&E staining. The results in Supplementary Fig. 14 revealed that the PMI and PTK hydrogels did not trigger obvious inflammatory reactions, and no inflammation cell accumulation was detected, which should be safe for in vivo applications.

## In vitro anti-oxidation, anti-inflammation, anti-bacteria, and cell homing

Inflammation and oxidative stress are two main pathological signs of progression in OA[39]. We first studied the antioxidative capability of the PTK hydrogel by 2,2-diphenyl-1-picrylhydrazyl (DPPH) free radical scavenging test. As presented in Fig. 3d, the DPPH scavenging efficiency by PTK hydrogel showed a concentration-dependent manner and more than 84% of DPPH was scavenged in a rather low concentration (1 mg/mL) of the hydrogel, which was due to the presence of catechol groups in TA[30,40–44]. However, the anti-oxidation property of the PMI hydrogel was relatively low (5.12%). The lipopolysaccharide (LPS) induced inflammation test was employed for anti-inflammation evaluation. The expression of inflammatory cytokine, interleukin-6 (IL-6), was similar in the blank, PMI, and PKG-treated groups 4 h after stimulation (Fig. 3e) and was significantly reduced by 5 times in the PTA and PTK hydrogel treated groups. And there was no obvious difference found between TA alone and the TA-included hydrogels, indicating the high anti-inflammatory efficiency of the PTK hydrogel.

Since surgical intervention may be accompanied by the risk of infection, the potent anti-bacterial capability should be owed to the implanted materials. Then the antibacterial efficiency of the PTK hydrogel was tested by the bacterial colony formation unit (CFU) counting method. As shown in Fig. 3f, after incubation with *Escherichia coli* (*E. coli*) (gram-negative bacterium) and *Staphylococcus aureus* (*S. aureus*) (gram-positive bacterium) for 12 h, the CFUs in TA (5 mg/mL), PTK, and PTA hydrogel-treated groups (10 mg/mL) were less than those in the KGN (100 mmol/L), blank and PMI-treated groups, which was attributed to the strong antimicrobial properties of TA. The quantitative results demonstrated that 93% and 98% killing efficiencies were achieved against *E. coli* and *S. aureus* by the PTK hydrogel (Fig. 3g), respectively. Meanwhile, TA and PTA-treated groups exhibited similar antibacterial properties. SEM observation and Live/Dead staining were performed to further evaluate the anti-bacterial capability of the hydrogels. The results in Supplementary Fig. 15 show that both *E. coli* and *S. aureus* lost their structural integrity after incubation with PTK hydrogel (10 mg/mL) and TA (5 mg/mL) for 12 h, and the bacterial structures remained intact in the control and PMI hydrogel treated groups, suggesting the promising anti-bacterial efficiency of the PTK hydrogel. Live/Dead staining results in Supplementary Fig. 16 revealed that KGN, PMI, and PKG hydrogels did not show antibacterial effects with strong green fluorescence (live bacteria). However, when the bacteria were treated with TA (5 mg/mL), PTA (10 mg/mL), and PTK (10 mg/mL) hydrogels, obvious red fluorescence (dead bacteria) was observed 12 h after incubation, which was attributed to the presence of TA in the hydrogel matrix[24,45].

Due to the limited regenerative capability of cartilage tissue, the reconstruction of cartilage in situ remains a considerable challenge. The current tissue engineering approaches for cartilage regeneration normally involve the encapsulation of BMSCs or other progenitor cells, which require additional isolation and culture processes in vitro[46]. Moreover, limited stem cell sources, potential ethical and safety concerns should also be considered for clinical applications. The cell homing strategy has been regarded as an alternative method for tissue

regeneration in recent years[47]. Various cytokines, such as TGF-β, SDF-1α, and functional peptides, were employed to recruit endogenous stem cells to the damaged area and induce directional differentiation to achieve cell-free cartilage regeneration[48,49]. However, the introduction of exogenous cytokines or peptides may cause potential immune reactions[21]. TA has been verified to feature the capability of cell migration on epithelial cells[24], renal endothelial cells[50], and bone marrow cells[26]. Therefore, we hypothesized that PTK hydrogel may induce BMSC homing in vivo for the regeneration of cartilage. The PTK hydrogel-induced BMSC migration behaviors were first investigated by scratch assay in vitro, and the original scratch areas were marked by red dashed lines (Fig. 4a). After 24 h of coincubation, there were more BMSCs migrated into the middle area in the PTA hydrogel, PTK hydrogel, and TA-treated groups than that in the other groups. The uncovered area was then identified and calculated to quantitatively analyze the capability for the promotion of cell migration (Supplementary Fig. 17). It was found in Fig. 4b that the uncovered areas in the PTK hydrogel, PTA hydrogel, and TA-treated groups were reduced by 3.8, 3.6, and 3.9 times compared to the control group, which were also significantly smaller than that those in the PMI and PKG hydrogel treated groups. Transwell assay was also adopted to investigate the BMSC homing ability of the PTK hydrogel as shown in Fig. 4c. BMSCs stained with crystal violet migrated through the chamber membrane in the PTK hydrogel, PTA hydrogel, and TA-treated groups were more than in the control group, PMI and PKG hydrogel treated groups. The semi-quantitative analysis indicated that the number of migrated cells in the PTK hydrogel, PTA hydrogel, and TA-treated groups was elevated by 6.4 to 6.9 times compared to that in the control group (Fig. 4d), and there was no significant difference detected in the control group, PMI and PKG hydrogel treated groups, suggesting that the promising BMSC homing capability was attributed to the presence of TA in the hydrogels. To investigate in vitro chondrogenic differentiation, the BMSC pellet culture test was performed. It was found that PTK hydrogel could induce larger cartilage pellet formation than that in the control group and PMI hydrogel treated group (Supplementary Fig. 18), suggesting the promising chondrogenesis property of PTK hydrogel. These results demonstrated that PTK hydrogel as a cell-free cartilage scaffold could induce the directional migration of BMSCs for further chondrocyte differentiation and cartilage regeneration.

## Macroscopic observation of in vivo cartilage regeneration

The in vivo cartilage regeneration mediated by PTK hydrogel was evaluated by surgery-induced cartilage defects in the SD rat model following previously published procedures[51]. Briefly, a full-thickness cartilage defect (2 mm in diameter and 2.0 mm in depth) was created in the center of the trochlear groove with an electric bone drill in both knees of 4-week-old SD rats. Different hydrogels were implanted into the right-side cartilage defects, and the rats were sacrificed at 8 weeks after surgery to observe the recovery of the created defects. As demonstrated in Fig. 5a, images of the femur condyles were taken and evaluated. In the control group, there was no cartilage tissue formation due to the limited regeneration capability of the native cartilage tissue. Since the PMI hydrogel lacked the on-demand bioactivities to promote cartilage tissue formation, the repair capability of the defects was neglectable. On the other hand, with the introduction of KGN or TA, the degeneration of cartilage tissue caused by the defects was partially inhibited by the PKG and PTA hydrogels and some calcific tissue and cartilage wear could be observed macroscopically. However, the uneven surface and clear boundaries of the regenerated tissues revealed their limited recovery capability. Notably, in the PTK hydrogel-treated group, the regenerated hyaline-like cartilage tissue with smooth surfaces exhibited intact connection with adjacent normal cartilage without obvious boundaries, which was similar to the normal cartilage tissue. The promising capability for cartilage regeneration may be attributed to

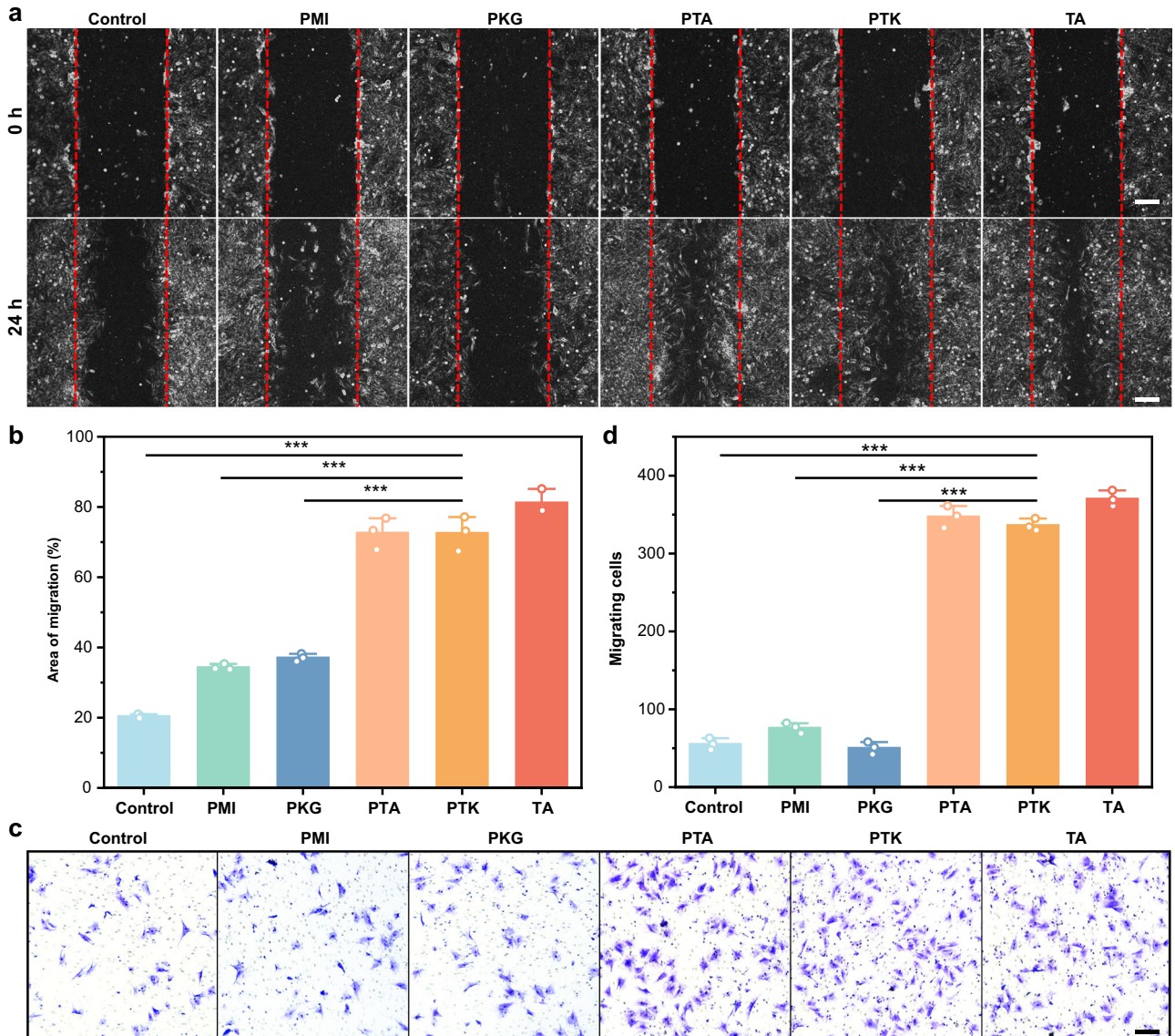

**Fig. 4 | In vitro cell migration test. a** Representative image of the migration behaviors of BMSCs with different treatments for 24 h. Scale bar: 200 μm. **b** Quantification of the relative migration areas in different groups. **c** Representative images of BMSCs after migration through a transwell (8μm, Corning) treated with hydrogel-conditioned DMEM or DMEM with TA for 24 h.

Scale bar: 50 μm. **d** Quantification of the migrated cell number based on the transwell assay. Values in b and d represent mean ± SD. One-way ANOVA test. Source data are provided as a Source data file. ($n = 3$ independent samples. *$P < 0.05$, **$P < 0.01$, ***$P < 0.001$).

the synergistic effect of TA and KGN for sequential BMSC homing and chondrocyte differentiation.

The International Cartilage Repair Society (ICRS) macroscopic scores were used to evaluate the repaired cartilage as depicted in Supplementary Fig. 19[51]. The PTK hydrogel-treated group showed the highest ICRS macroscopic score of 11.3 (the score was 12 for native cartilage), which was 2.9 and 5.7 times that in the PMI hydrogel-treated group and the control group, respectively. Comparatively, the ICRS macroscopic scores of the PTA and PKG hydrogel-treated groups were relatively lower (7.1 for PTA hydrogel and 7.9 for PKG hydrogel). Moreover, the weight-bearing ability of the legs treated with different hydrogels was also evaluated to reflect the functional recovery of the cartilage compared to the healthy legs (Supplementary Movie 1). It was found in Supplementary Fig. 20 that the interference of the hydrogels could improve the weight-bearing capability of the injured legs, and the rats treated with PTK hydrogel could bear a similar weight (98.1%) to the healthy rats, indicating its best efficiency for limb function recovery. The body weight of

the rats in all groups was monitored during the treatment. A decline in body weight was detected in the first week after surgery due to OA-induced pain (Supplementary Fig. 21), and body weight showed a normal increasing trend in the rest of the experimental period. We found that the body weight of the rats in the PTK hydrogel-treated group was the heaviest among all the groups, which also reflected its positive effect on cartilage functional recovery. These results demonstrated the superior cartilage regeneration capability of the PTK hydrogel.

### Histological and immunohistochemical evaluations, reverse transcription-polymerase chain reaction (RT-PCR), and western blot analysis

Histological and immunohistochemical staining were then employed to assess the cartilage regeneration ability in different groups, including H&E staining, toluidine blue staining, Safranin O/Fast Green staining, and immunohistochemical staining. For H&E staining, obvious cartilage defects were observed in the control group, and fibrous-like tissue

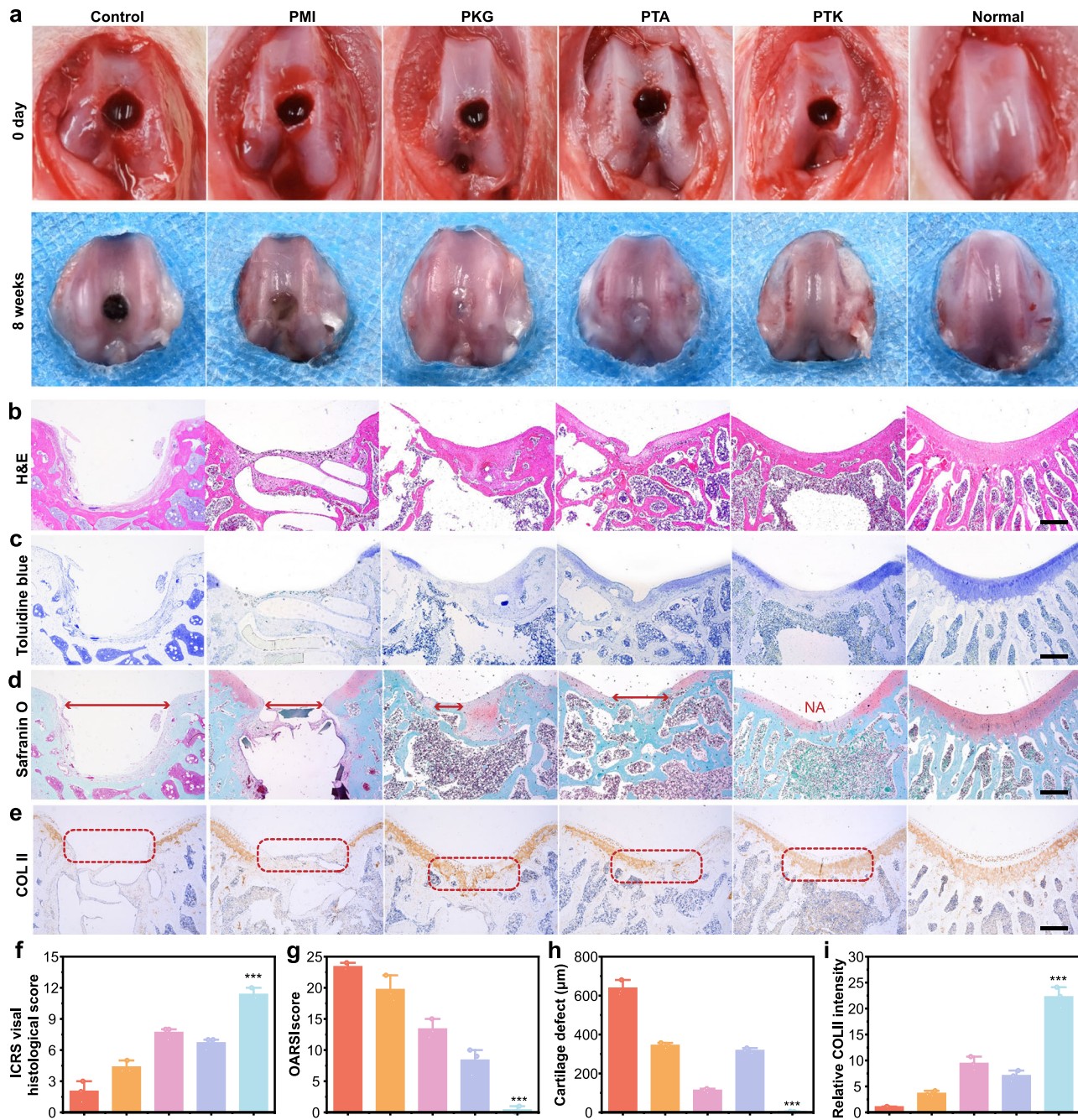

**Fig. 5 | Macroscopic, histological, and immunohistochemical evaluations.**
**a** Gross observation of the rat articular specimens at 0 and 8 weeks after operation with different treatments. Scale bar: 2 mm. Representative images of H&E staining (**b**), toluidine blue staining (**c**), Safranin O/Fast Green staining (**d**), and immuno-histochemical staining of COL II (**e**) in the control group, PMI, PKG, PTA, and PTK hydrogel treated groups. Scale bar: 200 μm. **f** ICRS histological scoring of the generated tissues in different groups. **g** OARSI histological scoring of the generated tissues in different groups. **h** The distances of cartilage defect in different groups.

**I** The relative intensities of COL II in different groups (The COL II expression level was calculated by Image J. The expression level of the control group was marked as $E_c$ and the COL II expression level of hydrogel treated group was marked as $E_h$. The relative COL intensity was calculated as follows: Relative COL II intensity = $E_h/E_c$). Values in **f**–**I** represent mean ± SD. One-way ANOVA test. Source data are provided as a Source data file. ($n = 3$ biologically independent samples. $*P < 0.05$, $**P < 0.01$, $***P < 0.001$).

formation was detected in the PMI hydrogel-treated group (Fig. 5b), suggesting their limited cartilage repair ability. For the defects treated with PKG and PTA hydrogels, although cartilage-like tissues were formed, the generated tissues were discontinuous with lacunae and nodose structures. Since TA and KGN could regulate in situ inflammation reactions and induce BMSCs to differentiate into chondrocytes, respectively, the ability to repair cartilage defects was improved. Notably, the cartilage tissues formed in the PTK hydrogel-treated group were

continuous and smooth with thicknesses similar to those of normal joint cartilage tissue. The same trend was also found in the results of toluidine blue staining (Fig. 5c), in which the proteoglycan in the extracellular matrix of cartilage was stained light blue. The areas stained with blue were significantly larger mediated by PTK hydrogel than those in the other groups. To further identify the neocartilage from bone tissue, Safranin O/Fast Green staining was conducted, by which the red-stained basophilic cartilage tissue can be distinguished from green-dyed

acidophilic bone tissue. As depicted in Fig. 5d, the thickness of the neocartilage was similar to that of normal tissue with red staining and the boundary between cartilage and bone was distinct in the PTK hydrogel-treated group, indicating efficient cartilage regeneration and prevention of cartilage calcification. Although TA could promote the migration of BMSCs into the defect areas, the homing cells could not differentiate into chondrocytes without further induction stimuli[6]. Thus, obvious calcific tissues with green staining were found in the PTA hydrogel-treated group. A limited capability for cartilage regeneration was also found in the PKG hydrogel-treated group due to insufficient stem cell homing. Furthermore, we performed immunohistochemical staining of alkaline phosphatase (ALP) and ELISA tests of ALP and Type I collagen (COL I) expression in different groups. As shown in Supplementary Fig. 22, there was no obvious difference in the expression level of ALP and COL I by immunohistochemical staining and ELISA tests for the healthy cartilage tissue and PTK hydrogel-induced new cartilage tissue. Type II collagen (COL II) is one of the major components of the cartilage extracellular matrix (ECM), and the distribution of COL II could reflect neocartilage formation directly in the defect area. As presented in Fig. 5e, there was negligible COL II staining in the control group and PMI hydrogel-treated group, and increased expression of COL II was detected when TA or KGN was introduced into the hydrogel matrix. Obviously, in the PTK hydrogel-treated group, the positive area for COL II staining was continuous with high intensity.

Moreover, the ICRS visual histological scores[51], osteoarthritis cartilage histopathology assessment system (OARSI)[52], hyaline cartilage defect distance, and relative expression of COL II were quantitively evaluated based on the results of H&E staining, toluidine blue staining, Safranin O/Fast Green staining, and COL II immunohistochemical staining, respectively. As shown in Fig. 5f, g, the highest ICRS score (11.3) and the lowest OARSI score (0.6) were achieved by the PTK hydrogel. No obvious cartilage fracture was detected in the PTK hydrogel-treated group (Fig. 5h), and the relative expression of COL II was enhanced by 22.1, 6.5, 2.3, and 3.2 times compared to the control group, PMI, PKG, and PTA hydrogel treated groups, respectively (Fig. 5I). These results also verified the promising cartilage recovery capability of the PTK hydrogel.

Reverse transcription-polymerase chain reaction (RT-PCR) and western blot analysis were employed to evaluate the chondrogenic markers in the generated tissues at the molecular level. The RT-PCR results in Fig. 6a–c show that the expression of chondrogenic genes (*ACAN*, *COL II*, and *SOX9*) in the joint tissue mediated by PTK hydrogel was 59.7, 13.3, and 4.7 times higher than that in the control group, which was also significantly higher than that in the PMI treated group. Moreover, *COL II* and *SOX9* expression in the PKG hydrogel-treated group were elevated by 3.6 and 3.5-fold compared to the control group, respectively, owing to the chondrogenesis property of KGN. Western blot analysis further revealed that the PTK hydrogel displayed the most positive effects on the expression of ACAN, COL II, and SOX9 (Fig. 6d). And the semi-quantitative results showed that the expression of ACAN, COL II, and SOX9 was enhanced by 52.6, 2.7, and 3.8 times, respectively, compared to that in the control group (Fig. 6e–g).

## Discussion

In this study, we fabricated a hydrogen-bond crosslinked and ultra-durable hydrogel (PTK hydrogel) loaded with TA and KGN for cell-free cartilage regeneration. Owing to the multiple hydrogen bonds in the network, the PTK hydrogel exhibited fast shape memory capability and could potentially be adopted for minimally invasive surgery applications. The mechanical properties of the implanted scaffold impose a significant impact on stem cell behaviors during cartilage matrix formation. As a result, the mechanical properties of the scaffolds could influence the differentiation, migration, and proliferation of adjacent cells[53]. However, the mechanical properties of previously reported hydrogels for chondrogenesis based on natural or synthetic polymers, such as alginates, chitosan, or polyethylene glycols (PEG), were considerably lower than those of natural cartilage tissue[54–56]. In this work, we demonstrated that the PTK hydrogel featured mechanical properties similar to those of natural cartilage tissue and could withstand 28000 loading-unloading mechanical cycles without fracture. Moreover, adequate tissue adhesiveness is beneficial to the stable fixation of implanted materials and enables long-term retention to construct a favorable microenvironment for tissue regeneration[35,57,58]. Owing to abundant catechol groups and multiple hydrogen bond formation, an ideal tissue adhesive property was also achieved in PTK hydrogel[30,59].

Articular damage is commonly associated with arthrostenosis in osteoarthritis patients who suffer from inflammatory joint adhesion. Traditional open surgery may cause physiological and psychological iatrogenic injury. Meanwhile, some joint areas are closely related to aesthetic or vital signs, such as the temporomandibular joint and

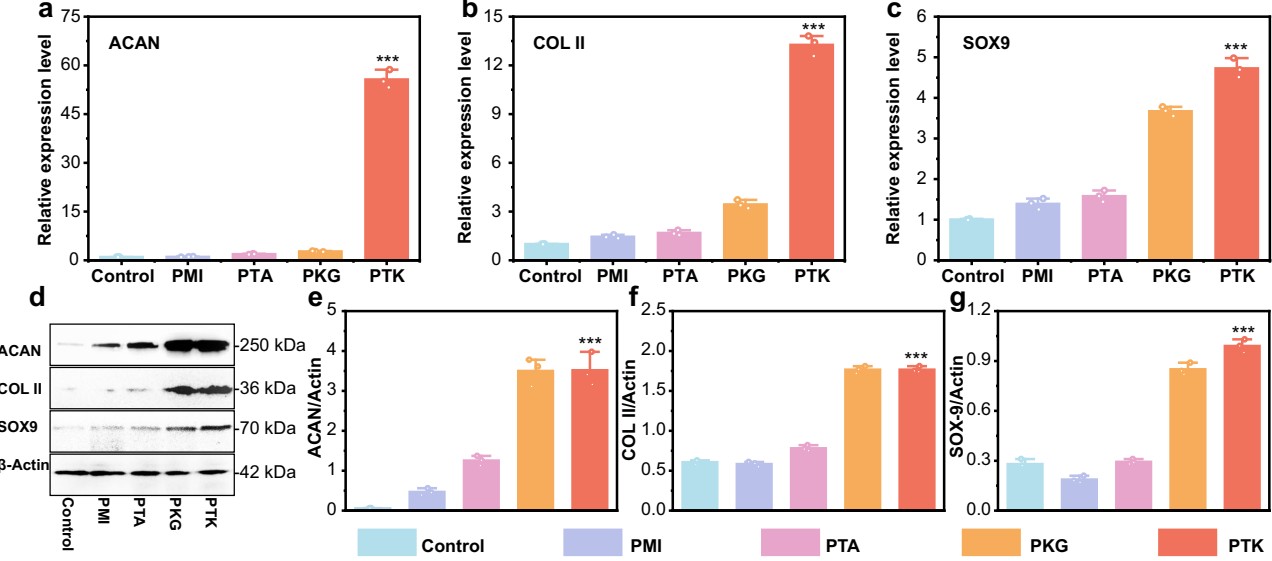

**Fig. 6 | RT-PCR and Western blot analysis.** The cartilage-related gene expression (ACAN (**a**), COL II (**b**), and SOX9 (**c**)) 8 weeks after surgery evaluated by RT-PCR. **d** Western blot analysis of ACAN, COL II, and SOX9. Semi-quantitative evaluation of ACAN (**e**), COL II (**f**), and SOX9 (**g**). Values in **a**–**c**, **e**, **f**, and **g** represent mean ± SD. One-way ANOVA test. Source data are provided as a Source data file. (*n* = 3 biologically independent samples. *$P < 0.05$, **$P < 0.01$, ***$P < 0.001$).

Cruveilhier joint, which are not appropriate for open surgeries. Minimally invasive surgery establishes an alternative approach to surmount these limitations by operating through small incisions and inserting surgical instruments using thin and flexible tubes[60]. Profiting from the dynamic hydrogen bond crosslinking network, the thermo-triggered fast shape memory properties were also achieved in the PTK hydrogel which could provide a useful solution for minimally invasive surgery. However, the fast recovery process may induce clogging during practical applications. Hydrogen bond-mediated shape memory is dependent on temperature, and a higher temperature could lead to faster shape recovery. Therefore, our further work will aim to improve the hydrogen bonding interactions to elevate the transition temperature of the network restructuring and ultimately control the shape recovery rate, by which the scaffold should be stable during endoscopy transfer and recover to the initial shape in an appropriate duration at the injury site.

Inflammation and oxidative stress are two major stimuli in OA development[39]. Previous studies have generally focused on the transplantation of stem cells or cartilage tissue[13,61]. However, the over-reacted inflammatory response and ROS release could lead to cell death, and thus, the regulation of the inflammatory reaction and clearance of over-expressed ROS should be on demand for cartilage regeneration. In vitro studies showed that the burst release of TA in the initial state could reduce the inflammatory reaction, clear the over-expressed ROS, and eventually construct a proper microenvironment for BMSC habitation. Benefiting from the stage-dependent drug release of TA and KGN, the PTK hydrogel could sequentially induce the migration of BMSCs into the hydrogel matrix and trigger directional differentiation to promote neocartilage formation. In vivo animal studies demonstrated that PTK hydrogel could efficiently promote the metabolic activity in the defect with similar properties as native tissues macroscopically and histologically.

Although promising cartilage regeneration was achieved with the treatment of the PTK hydrogel in this study, there are still several deficiencies we need to claim. First, approximately 40% of TA was released in the initial 4 h, and burst release may not enable the PTK hydrogel to efficiently exert its cell homing capability. Thus, the proper release rate of TA from the hydrogel matrix should be well considered. Moreover, due to the limited volume of bone joints, minimal invasion surgery by endoscopy to treat cartilage defect in rats is difficult to perform. Therefore, further studies need to be conducted on larger animals (pigs and dogs) to verify the advantage of the fast shape memory ability of PTK hydrogel. Given the complexity in shape and depth of different cartilage defects, the 3D print of hydrogel mimicking the cartilage structure should be more promising for cartilage engineering. We found that the injection of PMI or PKG solution in DMF into TA solution could form hydrogels, but the dimensions were not stable. In addition, TA may be oxidized at high temperatures when fused deposition modeling (FDM) is used, which could affect the bioactivity of the hydrogels. We will make great efforts to find suitable 3D-printing methods to mimic the sophisticated structures of native cartilage in the future.

## Methods

### Ethics statement
All animal experimental protocols were approved by the Animal Experimentation Ethics Committee of Xi'an Jiaotong University (2021-1081). The animals were kept in the SPF barrier. For euthanasia, $CO_2$ (1.5–2 L/min) was pumped at a constant rate into sealed cages to sacrifice the rats. The cervical dislocation was used to confirm death of experiment animals.

### Materials
PEG ($M_n = 6000 \text{ g mol}^{-1}$), MDI, IU, DBTDL, DMF, DPPH (2,2-diphenyl-1-picrylhydrazyl) and TA were purchased from Aladdin (Shanghai, China). Sodium hydroxide, ethanol and glutaraldehyde were purchased from Sinopharm Chemical Reagent Co., Ltd (Shanghai, China).

Deionized water was produced by ultra-pure water purification system from Beiyan Science and Technology Development Co., Ltd (Chongqing, China). KGN was synthesized according to the previous work[6]. Antibiotics, low glucose Dulbecco's modified Eagle's medium (LG-DMEM), and fetal bovine serum (FBS) were bought from Gibco (Grand Island, NY, USA). Calcein/PI cell viability/cytotoxicity assay (Live/Dead assay) kit, MTT cell proliferation and cytotoxicity assay kit and crystal violet staining solution were provided by Beyotime Biotechnology (China). LPS was purchased from Sigma-Aldrich. ALP, COL I and IL-6 enzyme-linked immunosorbent assay (ELISA) kit was purchased from Finetest (Wuhan, China) and Multi Science (Zhejiang, China), respectively. Anti-SOX9 antibody and Anti-Collagen II antibody were purchased from Abcam (UK). Anti-Aggrecan antibody was purchased from Proteintech (Wuhan, Hubei, China). Anti-beta Actin antibody was purchased from Boster Biological Technology Co., Ltd. (Wuhan, China). Live/dead BacLight™ Bacterial Viability Kit was purchased from Molecular Probes (Eugene, USA). Mesenchymal Stem Cell Chondrogenic Differentiation Medium was purchased from HyCyte™ (Suzhou, China). All chemicals were used as received.

### Fabrication and characterizations of PTK hydrogel
The PMI was synthesized through polycondensation of PEG, MDI, and IU in the presence of DBTDL. Afterward, KGN (2 mg/mL) was dispersed in PMI solution (20 mL) under ultrasonic agitation. The mixed solution was evaporated at 70 °C to obtain the PKG dry film. The PKG dry film was then immersed in 5% TA solution to obtain the PTK hydrogel. The FT-IR spectra of TA, KGN, PTA, and PTK hydrogels were recorded using FT-IR spectrometer (Thermo Fisher, Nicolet 6700). The microscopic morphologies of the hydrogels were characterized by scanning electron microscopy (SEM) (Hitachi, SU3500). Before measurements, the hydrogel samples were first lyophilized and sprayed with gold. During the preparation of freeze-dried samples, the hydrogel was first quickly placed into liquid nitrogen for 4 min, and then freeze-dried for 48 h to remove water.

### Shape memory property of PTK hydrogel
PTK hydrogel was cut into a strip membrane and crimped at 4 °C for 10 min to fix the temporary shape. The shape-recovery processes of PTK hydrogels under physiological conditions (37 °C, 95% humidity) were recorded. The recovery process and recovery rate were analyzed accordingly.

### Mechanical properties and adhesive strength of PMI, PKG, PTA, and PTK hydrogels
The mechanical properties of PMI, PKG, PTA, and PTK hydrogels were tested on CMT-1503 electromechanical tester (SUST Inc., China) at room temperature. All the samples (length = 16 mm, width = 4 mm, thick = 1 mm) were tested under 100% air humidity to prevent water evaporation during the examination. The stretching rate of the tensile test was set as 50 mm min$^{-1}$ and the stress-strain curve was obtained accordingly. The anti-fatigue capability was tested by consecutive loading in strain between 0% and 100%.

Fresh cartilage tissues were chosen as substrates for the lap shear test. The fresh pig costal cartilage was cut into a rectangular bar (15 mm × 40 mm) before washing with 1 M sodium hydroxide 3 times and rinsed in deionized water to remove adipose tissue. Then rectangular-shaped PTK and PTA hydrogels (1 cm × 1 cm) were applied to the surface of the cartilage tissues (Supplementary Fig. 7). The lap-shear test was tested on CMT-1503 electromechanical tester (SUST Inc., China). The adhesion strength of PTK hydrogel was measured at a cross-head speed of 50 mm min$^{-1}$. Each group was repeated 3 times.

### In vitro evaluation of drug-release behaviors of PTK hydrogel
The PTK hydrogel was cut into a round-shaped membrane (8 mm in diameter, 1 mm thick) and submerged with 40 mL PBS solution under

constant shake (60 rpm, 37 °C). The release solution was exchanged every 24 h and the TA and KGN release properties of PTK hydrogel were assessed by a UV–Vis spectrophotometer (Lambda 35, PerkinElmer) and HPLC system (SPD-M40, Shimadzu, Japan) independently. Briefly, for UV–Vis spectrophotometer evaluation, the UV-vis spectra of tannic acid (TA) in PBS were recorded using UV–Vis spectrophotometer to obtain the standard absorption curve of the TA solution (276 nm). The KGN release was determined by HPLC system with a WondaSil C18-WR column (250 × 4.6 mm, pore size 5 μm, Shimadzu, Japan). The effluent was detected with a UV detector at 270 nm for KGN.

### Biocompatibility evaluation

The cytocompatibility of the hydrogels was evaluated by Live-Dead staining and MTT assay. Briefly, BMSCs were isolated from the bone marrow of 4-week-old SD rat. Both femurs were separated after the rat was gently sacrificed. The bone marrow was flushed out by LG-DMEM with a 20-gauge needle attached to a 2 mL syringe. The cell suspension was filtered with 70 μm cell strainer (Millipore, Sigma-Aldrich, USA) and cultured in complete LG-DMEM (37 °C, 5% carbon dioxide, 95% humidity). The cells after 3-4 passages were used in the following experiments[20,62]. Live/Dead staining was conducted 24 and 96 h after incubation of BMSCs with hydrogel extractions (50 mg/mL). The cell viability was observed under a fluorescent microscope (DMi8, Leica, Germany) after staining. MTT assay was conducted on BMSCs according to the published protocol[63].

The in vivo tissue compatibility was evaluated by subcutaneous implantation of the hydrogels (8 mm in diameter, 2 mm in thickness, 100 mg). The local inflammatory reaction was evaluated after 7 and 30 days of implantation by H&E staining. The long-term in vivo biocompatibility was evaluated by histological observation of the main organs of rats two months after hydrogel implantation.

### In vitro antibacterial property of TAP hydrogels

The antibacterial activity of hydrogels against *E. coli* and *S. aureus* was assessed using an agar disc diffusion test. *S. aureus* and *E. coli* were cultured in Luria-Bertani (LB) broth at 37 °C overnight. TA (5 mg/mL), KGN (100 mmol/L), PMI, PTA, PKG, and PTK hydrogels (10 mg/mL) were added into 5 mL bacterial suspension containing $10^6$ CFU $mL^{-1}$ *E. coli* or *S. aureus* and cultivated at 37 °C for 12 h, respectively. The bacterial suspension of each group (100 μL) was spread onto the surface of the corresponding LB agar plate without further dilution. The plates were incubated for 12 h before CFU counting. Furthermore, a Live/Dead BacLight Bacterial Viability Kit (Molecular Probes, Eugene, OR, USA) was used. Briefly, TA (5 mg/mL), KGN (100 mmol/L), PMI, PTA, PKG, and PTK hydrogels (10 mg/mL) were added into 5 mL bacterial suspension containing $10^6$ CFU $mL^{-1}$ of *E. coli* or *S. aureus*, which were cultivated at 37 °C for 12 h, respectively. Then, SYTO 9 (1.5 μL/mL) and propidium iodide (1.5 μL/mL) of Live/dead BacLight™ Bacterial Viability Kit were added to the samples. Bacteria viability was then microscopically observed (fluorescent microscope (DMi8, Leica)). The morphologies of *E. coli* and *S. aureus* were examined by SEM observation (SU3500, HITACHI, Japan). Briefly, *E. coli* and *S. aureus* were incubated with TA (5 mg/mL), PMI, and PTK hydrogels (10 mg/mL) for 12 h. Then, *E. coli* and *S. aureus* were fixed with 2.5% glutaraldehyde in sodium phosphate buffer (pH 7.4) for 1 h. Subsequently, *E. coli* and *S. aureus* were rinsed with sodium phosphate buffer and dehydrated with a graded ethanol series. SEM images were taken at an acceleration voltage of 5 kV.

### Antioxidant property evaluation

The hydrogel was freeze-dried and cut into fine pieces before being ground into powder. Afterward, different amounts of the hydrogel powder samples were dispersed in 5 mL 100 μM DPPH (2,2-diphenyl-1-picrylhydrazyl) ethyl alcohol solution to form different concentrations of PTK (0.5, 1, 2, 3, 5, 10 mg/mL) and PMI (10 mg/mL) solutions. The

mixture was stirred and incubated in a dark place at 37 °C for half an hour. Then, the UV absorption at 517 nm was scanned (UV–Vis spectrophotometer (Lambda 35, PerkinElmer)). The UV absorption of the initial DPPH solution was marked as $A_i$. The absorption after incubation was marked as $A_f$. The degradation of DPPH was calculated by the following equation:

$$DPPH\ scavenging\% = (A_i - A_f)/A_i \times 100\% \quad (1)$$

### Anti-inflammation assay

10 μg $mL^{-1}$ LPS treated BMSCs were seeded at a density of $2 \times 10^5$ cells per well to 6-well plate and cultured overnight. The cells were subsequently treated with different hydrogel extracts with 50 mg $mL^{-1}$. After 24 h stimulation, culture supernatants were collected. IL-6 concentration in the supernatants was measured by enzyme-linked immunosorbent assay (ELISA; Multi Sciences Biotech, Co., Ltd, Shanghai, China) according to the manufacturer's directions.

### Cell migration assay

The in vitro scratch assay and transwell assay were used to study the migration of BMSCs. In the in vitro scratch assay, the cell distribution images were acquired at 0 and 24 h after stimulation, and the scratch areas were quantified by Image J 1.52 software. The initial area was marked as $S_i$ and the residual area was marked as $S_r$. The rate of migration was calculated as follows:

$$Migration\ rate(100\%) = (S_r - S_i)/S_i \times 100\% \quad (2)$$

For the transwell assay, after incubation at 37 °C for 24 h, the migrated cells were stained with 1% crystal violet for observation and quantitative analysis.

### In vitro chondrogenic differentiation assay

For chondrogenic differentiation in high-density pellet cultures, $2.5 \times 10^5$ of BMSCs were centrifuged to form a pellet. PMI and PTK hydrogels (10 mg/mL) were incubated with defined serum-free chondrogenic medium for 12 h, which was used to co-culture with the pellets for 21 days. Control pellets were cultured in the same medium without hydrogel treatment. The culture medium was changed three times a week. The pellet sections were then stained with alcian blue for microscopic observation.

### In vivo evaluation of cartilage regeneration

The acute surgical OA model of SD rats (200–250 g, 4 weeks old) was used for the in vivo study. The acute surgical cartilage defect model of SD rats was used for the in vivo study. A full-thickness cartilage defect was created in the center of the trochlear groove with an electric bone drill in both knees of the rats. The round-shaped PMI, PTA, PKG, and PTK hydrogels films (2 mm in diameter, 2 mm in thickness, 6.25 mg) were filled into the cartilage defects in the right joint and the left knees were untreated as control. The PTA and PTK hydrogels contained 2.2 mg of TA. The PTK hydrogel contained approximately 0.008 mg of KGN. The recovery of the cartilage defects mediated by different hydrogels was assessed by gross behavior observation, histological and immunohistochemical evaluations, RT-PCR, and western blot analysis.

The rehabilitation of surgery-induced joint dysfunction was evaluated through load capacitance measurements. The weight-bearing capacity of lower limbs was assessed by the load-bearing test in which the maximum weight-lifting ability before and 8 weeks after the surgery were recorded (Movie S1). The percentage of load-bearing was

calculated as follows:

$$Load - bearing(\%) = \frac{(initial\ strength - recovered\ strength)}{initial\ strength} \times 100\%$$

$$(3)$$

### Quantitative real-time polymerase chain reaction (RT-PCR)

The cartilage-related gene expression in all groups was investigated by RT-PCR. The joint cartilage samples (all the cartilage on the distal femur) were cut into fine pieces and homogenized after 8 weeks of treatment. The same amount of total RNA (1 µg) was reverse-transcribed using a First Strand cDNA Synthesis Kit (Fermentas) according to the manufacturer's instruction. IQ5 kit (Bio-Rad, Hercules, CA, USA) was used to perform the RT-PCR test. The sequences of the primers were listed in Table S1.

### Western blot analysis

The collected joint samples were homogenized in RIPA buffer (Sigma) and supplemented with protease and phosphatase inhibitors (ThermoFisher). A total of 60 µg of proteins from each group were loaded and fractionated electrophoretically on SDS-PAGE gel (Beyotime, Cat. #P0012AC). The target proteins on the SDS-PAGE gel were transferred to polyvinylidene fluoride (PVDF) membranes (Merck Millipore, Cat. #HY-IPVH00010) and blocked with 5% blocking buffer at 37 °C for 60 min. Subsequently, the membranes were incubated with primary antibodies against SOX-9 (1:2000 dilution, Abcam, Cat. #Ab185966, Clone: EPR14335-78), COL II (1:1000 dilution, Abcam, Cat. #Ab188570, Clone: EPR12268), ACAN (1:1000 dilution, ProteinTech, Cat. #13880-1-AP, RRID: AB_2722780), and β-actin (1:5000 dilution, BOSTER, Cat. #BM3873, Uniprot IDACTB: P60709) at 4 °C overnight, and were washed three times by TBST (Sangon Biotech Co., Ltd. Shanghai, China). Then, the membranes were incubated with horseradish peroxidase-conjugated secondary antibodies (1:200 dilution, BOSTER, Cat.# BA1054) for 2 h at room temperature. The immunoreactive protein bands were detected with electrochemiluminescence (ECL) reagent (Millipore, Cat:# WBKLS0100) and acquired by ChemiDoc TM Touch Imaging System (BIORAD, Hercules, CA, USA).

### Statistical analysis and reproducibility

Statistical analysis was performed using SPSS (version 2020.0.0). The initial sample size was estimated by previously published experiment protocols[47]. Sample size or biological replicate was presented in the figure legends. For each result shown by representative images, the stability of the results has been verified by at least three independent experiments. The One-way ANOVA test was performed to identify the correlation between the dependent variables. $P$ value less than 0.05 was considered statistically significant.

### Reporting summary

Further information on research design is available in the Nature Portfolio Reporting Summary linked to this article.

## Data availability

All data supporting the findings of this study are available within the article and its supplementary files. Any additional requests for information can be directed to, and will be fulfilled by, the corresponding authors. Source data are provided with this paper.

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

## Acknowledgements

This work was financially supported by the National Natural Science Foundation of China (NSFC 52322309 and 52173139), the "Young Talent Support Plan" of Xi'an Jiaotong University, and Fundamental Research Funds for the Central Universities (xzy022021040, xzy022021052, xzy012023098, and xzy012023105). The authors gratefully acknowledge Mr. Zijun Ren, Ms. Yanan Chen, and Ms. Lu Bai at Instrument Analysis Center of Xi'an Jiaotong University for their assistance with SEM and UV tests.

## Author contributions

Y. C. and Y. Y. conceived the project and designed the experiments. Y. Y. conducted animal experiments and data analysis. X. Z contributed to the in vitro cytocompatibility evaluation. Y. Z., S. W., A. Y., X. Z. and X. C. participated in the revision of the manuscript. Y. C. supervised the whole work and wrote the manuscript. All authors commented on the manuscript.

## Competing interests

The authors declare no competing interests.
