## [Peer review file · Nature Communications]

REVIEWER COMMENTS

Reviewer #1 (Remarks to the Author):

The authors present a hydrogel for cartilage repair and performed a variety of tests to characterize their properties and healing capabilities. They present evidence of the ability of the hydrogels to reduce inflammation, promote MSC migration and support cartilage formation using in vitro and in vivo assays. While the work is interesting, several areas need to be addressed.

In general, the manuscript needs to be read by an editor. There are numerous grammatical errors.

There are several instances where the methods require clarification, especially in regards to the type and concentration of hydrogels used in the in vivo studies. These are highlighted in the annotated manuscript.

The proper methods for characterizing and reporting the mechanical properties of hydrogels were not used in these studies and should be re-examined by the authors. When proper mechanical tests were performed, the data was not analyzed to yield quantitative values other than fracture of the materials.

It is unclear how the test performed to assess changes in weight-bearing are associated with loading of the knee joint.

It is unclear how higher metabolic activity in the joints for 30 minutes reflects cartilage healing over the 4 weeks. The authors state that the higher levels of metabolism measured by PET is due to the chondrogenic activity of the cells and increased blood flow. Since this is a fairly new, non-standard way of assessing cartilage regeneration, a more thorough understanding of how the various cells in the joint may be contributing to the metabolic activity is required.

MSC-based cartilage formation may be fibrocartilaginous or the cells may exhibit markers indicating maturation down the endochondral ossification pathway. Therefore, non-cartilaginous markers should be investigated with immunoblotting and staining.

There are many instances where discussion is mixed in the results section. Several (but not all) of these are highlighted in the annotated manuscript. The discussion section only addressed the burst release of TA and the need for large animal models and is, thus, deficient.

Quite a few details on experimental design (e.g., treatment groups for in vivo studies) are relegated to the Supplementary section and should be moved to the main body of the manuscript.

Reviewer #2 (Remarks to the Author):

In the present work, the authors developed an ultra-durable cell-free bioactive hydrogel scaffold with fast shape memory and on-demand drug release for directional migration and differentiation of stem cells to promote cartilage regeneration. This hydrogel may be a new way to address the problem of cartilage regeneration. After carefully evaluating the manuscript, the manuscript in the present form will not be considered for publication in Nature Communications.

1. The experiments of antibacterial activity just investigated by spread plate method that was not enough. SEM and live/dead stain fluorescent test should be used to observe the morphology of the bacteria.
2. In vitro, the ability of PTK to recruit and promote differentiation of bone marrow mesenchymal stem cells to chondrocytes was verified.
3. All Figures in the manuscript are not clear enough, and the quality of the picture could be improved.
4. Please confirm the accuracy of the sequential labeling of the pictures in the manuscript. For example, the antibacterial content in line 265 should correspond to Fig. 3g, rather than Fig. 2g.
5. The long-term cytotoxicity should be evaluated for gels, which demonstrates whether the degradation products have low toxicity.
6. How can hydrogels dissolve into solutions? How do you make the concentration gradient difference? Hydrogel is a gelatinous substance. Please explain why mg/mL is used to express the content of hydrogel.
7. Please explain how to ensure the rapid shape memory action and the feasibility of manipulation of the material in room temperature within 30 seconds.
8. Does rapid thermo-triggered properties limit its practical application?
9. Antibacterial experiments should add TA group and KGN group to prove which the component has antibacterial effect.
10. In vitro stem cell experiments are needed to demonstrate whether the material induces cartilage tissue generation.

11. PET characterization method is novel, but what is the connection with PTK?
12. The topic of the article has to do with on-demand drug release, but there is little research on this content in the in vitro drug release section. The encapsulation and loading rates of the drug were not clearly studied, and the cause of TA's sudden release in a short time was not explained.
13. In the studies of anti-inflammatory, anti-oxidation and antibacterial effects of materials, the full manuscripts only emphasized the effect of the encapsulated drugs, but did not discuss the influence of the carrier on these aspects.

Reviewer #3 (Remarks to the Author):

The manuscript by Shuang Wang et al. is concerned with cell-free transplantable hydrogel scaffold for the cartilage regeneration by stem cell recruitment and its differentiation. In particular, the author presents the novel shape-memory drug-loaded hydrogel to facilitate cell migration and stem cell differentiation to chondrocyte. They employed tannic acid (TA) kartogenin (KGN) for multiple purposes to induce cartilage regeneration. They also show hydrogel-mediated cartilage regeneration using model organisms indicating the formation of neocartilage in vivo. This paper has great significance in that it presented hydrogen bond-based stage-dependent drug release from namely PTK hydrogel and consequently it induced homing of bone marrow stem cells followed by their differentiation; therefore this work would attract many researchers' attention to challenge current huddles to restore the defects on articular cartilages. However, there are a couple of points to be elucidated and confirmed. Suggestions regarding the manuscript are below:

1. PTK film was fabricated by drying after polycondensation of monomers followed by KGN and TA adsorption. This indicates the randomness of hydrogel structure in manufacturing. However there is a certain level of directionality in articular/hyaline cartilage structure, which is horizontally aligned chondrocytes in superficial zone and vertically aligned chondrocytes in middle/deep zone. Is it possible to 3D-print this hydrogel to render some level of aligned structure and to mimic structural, functional and mechanical characteristics of native cartilage tissue?
2. The author explained the expected function of TA in PTK hydrogel. Even though the author listed a couple of references for the feature of TA to induce cell migration in different cell types, the author still needs to show additional references for TA to regulate inflammation and to reduce oxidative stress.
3. The author presented the antibacterial capability by testing the bacterial colony formation unit (CFU) counting method. If the bactericidal effect came from TA, the author should also show a PTA-treated group which consists of hydrogel only with TA to claim antibacterial effect of TA.
4. What is the mechanism for TA burst release? If it is gradient-driven release from the hydrogel, the author would consider covalent linkage of TA with cleavable sequence to hydrogel monomer to control the release rate of TA from hydrogel after transplant.

5. The shape memory test conducted in this study is quite short. If this test was for the possibility of hydrogel transport through endoscope, the author should be able to control shape recovery rate to avoid clogging of hydrogel in the endoscope in the minimally invasive surgery.

6. The main driving force for the hydrogel to hold TA is the hydrogen bonds between polycondensated PMI and TA, therefore the release of TA from hydrogel can be affected by pH changes. Environmental pH is one of the crucial parameters that greatly affects enzyme activity and cellular biochemical reactions involving tissue repair and homeostasis. Author should conduct a TA release test in different ranges of pH to consider comparable conditions with articular cartilage defect microenvironment.

7. The hydrogel pore size and porosity measurement is important to characterize the BMSC homing effect of hydrogels. The author needs to measure the pore sizes and porosities of PMI, PKG, PTA and PTK. The author should also track the changes of pore size/porosity upon pH changes

Reviewer #4 (Remarks to the Author):

Note: This review is limited to aspects of the manuscript pertaining to the use of PET imaging.

Summary: This manuscript presents the development of a hydrogel scaffold to promote cartilage regeneration. As part of the study, the authors used Na¹⁸F PET imaging to image bone remodeling non-invasively in mouse models. NaF is a well-established tracer, used to measure bone remodeling in vivo, including bone metastases and musculoskeletal disorders. The use of Na¹⁸F to image cartilage regeneration seems logical and is novel. Specific comments are provided below:

1. Background:

- It would be helpful to provide additional background and rationale pertaining to the mechanism of deposition of Na¹⁸F during bone remodeling. ¹⁸F ion is chemically analogous to calcium and is incorporated into the bone matrix as fluoroapatite.

- The following statement is incorrect: "¹⁸F-sodium fluoride (Na¹⁸F) is the most commonly used PET tracer". The most common PET tracer is ¹⁸F-fluorodeoxyglucose, which is used to image glucose metabolism.

- The manuscript refers to Na¹⁸F uptake as a marker of metabolic activity, but this may not be exact since ¹⁸F ions bind to bone by chemisorption, therefore the uptake of the tracer depends primarily on the area of exposed bone surface (<https://www.ncbi.nlm.nih.gov/pmc/articles/PMC3169430/>).

2. Methods: The details of the imaging experiments are vague, and the following information should be provided:

- Whether CT images were also acquired
- Duration of each acquisition
- Where the tracer was obtained / produced
- Image reconstruction and any filtering applied
- How the regions of interest were drawn, and whether the analysis was performed on a 2D slice or on the full 3D volume
- Whether decay correction was applied

3. Findings: The results of the PET study are provided in Figure 5. Panel b-c shows the PET images whereas panels d-e shows the quantitation of the images.

- There seems to be a discordance between the images and the quantitative analysis. The analysis suggests a 9 fold difference between PTK hydrogel and control, however, the images do not show an increase of signal of that magnitude. At most there is maybe a 2x enhancement of the signal (based on the color scale). Furthermore, the increase is unlikely to be specific to the treatment since an increase is also observed in the untreated contralateral knee.
- Could the ROI analysis be applied to the contralateral knee as a control?
- The rainbow color scale should include quantitative units for reference.
- It is not clear if the images shown in b-c are maximum intensity projections or cross-sectional images. Also why is there no signal from the bladder? $^{18}\text{F-NaF}$ should have rapid renal clearance. Could additional orthogonal views be included to clarify where the area of interest (dated box) is located with respect to the body?
- The text should clarify how the "relative nuclide accumulation" was computed. Was the reference taken as the contralateral (healthy) knee from the same animals, or the damaged but untreated knee from the control animals? An internal reference should be used to normalize PET signal within each animal to account for possible variations in injected dose, etc.
- The 30 min superposition may not be the best way to analyze the dynamic data. Kinetic analysis using a compartment model would be more accurate and could help distinguish between tracer delivery effects (blood flow) and tracer deposition into bone matrix. If this is not possible, the image obtained at 30 min would be more relevant since the earlier images will mainly show blood perfusion.

4. Minor points:

L348 reflex  reflect

L369: ... dash  dashed line

We thank the reviewers for the constructive comments and suggestions for our manuscript entitled “*Ultra-durable Cell-free Bioactive Hydrogel Scaffold with Fast Shape Memory and On-demand Drug Release for Directional Migration and Differentiation of Stem Cells to Promote Cartilage Regeneration*” (NCOMMS-22-44392). We have carefully revised the manuscript based on your comments, and hopefully, our revision is satisfactory for publication in **Nature Communications**.

In addition to edits and clarifications to the text, the manuscript includes the following new data:

1. The comparison of mechanical properties between porcine cartilage and PTK hydrogel (Supplementary Fig. 5).
2. Immunohistochemical staining of ALP (alkaline phosphatase) in healthy rats and PTK hydrogel-treated rats with cartilage defect (Supplementary Fig. 22a and b).
3. ELISA test of ALP expression in the control group and PTK hydrogel-treated groups (Supplementary Fig. 22d).
4. In vitro cartilage differentiation of BMSCs induced by PMI and PTK hydrogels (Supplementary Fig. 18).
5. In vivo long-term biocompatibility of PMI, PKG, PTA, and PTK hydrogels (Supplementary Fig. 12).
6. In vitro antibacterial properties of TA, KGN, PMI, PKG, PTA, and PTK hydrogels (Fig. 3f).
7. Live/Dead staining of anti-bacterial properties of PMI, PKG, PTA, and PTK hydrogel as well as KGN and TA (Supplementary Fig. 16).
8. SEM images of *E. coli* and *S. aureus* with the treatment of TA, PMI, and PTK hydrogels (Supplementary Fig. 15).
9. The drug loading content (DLC) of PTK hydrogel.
10. In vitro drug release of PTK hydrogel at different pHs.
11. SEM images of PTK hydrogel at different pHs.

Point-by-point responses to the reviewers’ comments are noted below with major changes to the manuscript noted in red text.

Reviewer #1:

The authors present a hydrogel for cartilage repair and performed a variety of tests to characterize their properties and healing capabilities. They present evidence of the ability of the hydrogels to reduce inflammation, promote MSC migration and support cartilage formation using in vitro and in vivo assays. While the work is interesting, several areas need to be addressed.

1. In general, the manuscript needs to be read by an editor. There are numerous grammatical errors.

Our reply: *Thanks for your comments. We have carefully revised the manuscript and made corrections based on the reviewer's suggestion.*

2. There are several instances where the methods require clarification, especially in regards to the type and concentration of hydrogels used in the in vivo studies. These are highlighted in the annotated manuscript.

Our reply: *Thanks for your comments. The PMI (without drugs) and PTK (TA and KGN loaded hydrogel) hydrogels used in the in vivo tissue compatibility test were cut into the same shape (8 mm in diameter, 2 mm in thickness, 100 mg), and the in vivo degradation rate was monitored by the change of the weight of the residual hydrogel at predetermined time intervals. For the in vivo cartilage regeneration test, the implanted hydrogels were cut into the same size (2 mm in diameter, 2 mm in thickness, 6.3 mg), which could properly fill the cartilage defect made by the bone drill. The PTA and PTK hydrogel contained 2.2 mg of TA and PTK hydrogel contained approximately 0.008 mg of KGN based on the drug loading content test (as shown below). We have made clarifications in this experiment section.*

a) TA loading content in PTA and PTK hydrogels. b) KGN loading content in PKG and PTK hydrogels.

3. The proper methods for characterizing and reporting the mechanical properties of hydrogels were not used in these studies and should be re-examined by the authors. When proper mechanical tests were performed, the data was not analyzed to yield quantitative values other than fracture of the materials.

Our reply: *Thanks for your comments. We studied the tensile and compression properties of PTK, PMI, and natural porcine cartilage as shown below and*

Supplementary Fig. 5. The results showed that the introduction of TA improved the ultimate tensile stress of PTK hydrogel compared to PMI hydrogel, and the compression stress at the strain of 40% by PTK hydrogel was lower than that by the natural cartilage. Although the mechanical strength of PTK hydrogel was relatively low, it was still higher than those hydrogels reported in previous work for cartilage regeneration (*Biomaterials*. 2021. 279:121214; *Acta Biomater*. 2021. 1, 128:1-20.).

Supplementary Fig. 5 a) Tensile stress-strain curves of natural porcine cartilage, PMI and PTK hydrogels. **b)** Compress stress-strain curves of natural porcine cartilage, PMI and PTK hydrogels.

It was also found that the introduction of TA significantly increased the antifatigue property of PTK hydrogel in successive loading-unloading cycles (28000 cycles) while the PMI hydrogel broke down at approximately 1600 cycles. The change of the maximum stress and the retention ratio of the initial stress at 0, 20, 200, 1000, 3000, 6000, 10000, and 20000 cycles by PTK hydrogel were quantitatively analyzed (Supplementary Fig. 6). We found that the maximum tensile stress of the PTK hydrogel was constant for 20000 cycles, and more than 90% of the original value was retained (Fig. 2c). However, PMI hydrogel was broken after 1600 cycles, demonstrating the excellent cyclic stability and durability of PTK hydrogel. The formation of multiple hydrogen bonds in the network endowed PTK hydrogel with outstanding anti-fatigue properties. As the reviewer suggested, the antifatigue property of natural cartilage should be involved for comparison. However, the natural cartilage tends to be broken at approximately 40% tensile strain (*J Orthop Res*. 1986, 4(4):379-92). The antifatigue evaluation in this work was conducted between 0% and 100% strain, which was not applicable to natural cartilage. Moreover, as the reviewer suggested, the demonstration of weight lift and tissue adhesive by images was removed from the manuscript.

Supplementary Fig. 6 Maximum stress of PMI and PTK hydrogels at 0, 20, 200, 1000, 3000, 6000, 10000, and 20000 cycles during the successive loading-unloading test.

Fig. 2c Retention ratio of the initial stress at the 0, 20, 200, 1000, 3000, 6000, 10000, and 20000 cycles during the successive loading-unloading test for PMI and PTK hydrogels.

4. It is unclear how the test performed to assess changes in weight-bearing are associated with loading of the knee joint.

Our reply: Thanks for your comments. The weight-bearing test was commonly used in previous studies (*Science*. 2012. 11;336(6082):717-21; *Osteoarthritis Cartilage*. 2006. 14(10):1041-8; *Osteoarthritis Cartilage*. 2003. 11(11):821-30) to reflect the alleviation

of OA-induced pain, and we also added the experimental details in the revised Supplementary Information and Movie S1.

5. It is unclear how higher metabolic activity in the joints for 30 minutes reflects cartilage healing over the 4 weeks. The authors state that the higher levels of metabolism measured by PET is due to the chondrogenic activity of the cells and increased blood flow. Since this is a fairly new, non-standard way of assessing cartilage regeneration, a more thorough understanding of how the various cells in the joint may be contributing to metabolic activity is required.

Our reply: *Thanks for your comments. Quantitative Na¹⁸F-based PET/CT has been reported to be useful for the assessment of metabolic, degenerative, traumatic, and neoplastic bone diseases (Semin Nucl Med. 2015. 45(1): 58–65). The standardized uptake value of osteochondral composites depends on the blood flow to the cartilage defect area, exposed bone surface area, regional osteochondral activity, and renal clearance (J Nucl Med. 2012. 53(8):1175-1184.). The rapid focal uptake of Na¹⁸F occurs preferentially at sites with high osteo-chondrogenesis activity. Hence, the component of osteochondral turnover measured by ¹⁸F imaging is related to the chondrogenesis activity. Moreover, as the inflammation reaction was alleviated 4 weeks after surgery, the blood flow in different groups primarily reflected the cartilage regeneration activity. We have added the background in the revised Manuscript.*

6. MSC-based cartilage formation may be fibrocartilaginous or the cells may exhibit markers indicating maturation down the endochondral ossification pathway. Therefore, non-cartilaginous markers should be investigated with immunoblotting and staining.

Our reply: *Thanks for your comments. In our work, the cartilage calcification was evaluated by Safranin O/Fast Green staining (Fig 6C), in which basophilic cartilage tissue (proteoglycan) was stained in red by Safranin-O and ossification tissues such as bone and mineralized cartilage was stained in green by Fast Green (Biomaterials 2014, 35, 9984-9994; Biomaterials 2019, 210, 51-61; Adv. Funct. Mater. 2019, 29, 1807356; Adv. Mater. 2021, 33, 2008451). It was found that the regenerated cartilage tissues in the PTK hydrogel-treated group were continuous with red staining, and the boundary between cartilage and bone was distinct, suggesting the prevention of cartilage calcification. Furthermore, we performed immunohistochemical staining and ELISA test to investigate the expression of alkaline phosphatase (ALP) in different groups. As shown in Supplementary Fig. 22, there was no obvious difference in the expression level of ALP by immunohistochemical staining and ELISA test for the untreated healthy*

cartilage and PTK hydrogel-induced new cartilage tissue. We have added these results in the revised Manuscript and Supplementary Information.

Supplementary Fig. 22 a) Immunohistochemical staining of ALP in untreated healthy cartilage. b) Immunohistochemical staining of ALP in PTK hydrogel treated group. c) Semiquantitative analysis of ALP expression in untreated healthy cartilage and PTK hydrogel treated groups based on immunohistochemical staining. d) ELISA test of ALP expression in untreated healthy cartilage and PTK hydrogel treated group.

7. There are many instances where discussion is mixed in the results section. Several (but not all) of these are highlighted in the annotated manuscript. The discussion section only addressed the burst release of TA and the need for large animal models and is, thus, deficient.

Our reply: Thanks for your comments. As an implanted hydrogel designed for cartilage regeneration, the fast shape memory process of PTK hydrogel may be not beneficial to the transfer process by the endoscope. To address this issue, further study should be focused on the following two aspects: 1. The shape recovery speed is dependent on temperature, and a higher temperature leads to faster shape recovery. Thus, the material design should aim to improve the hydrogen bonding interactions to elevate the transition temperature of the network restructuring, by which the scaffold should be stable during endoscopy transfer and recover to the initial shape in an appropriate duration in the injury site. 2. The design of heat insulation coating on the endoscope may be an alternative way to avoid undesired shape recovery during operation.

Moreover, given the complexity in shape and depth of different cartilage defects, the 3D printing of hydrogels to mimic the cartilage structure should be more promising for

cartilage engineering. We found that the injection of PMI or PKG solution in DMF to TA solution could form hydrogels, but the dimension was not stable. In addition, TA may be oxidized at high temperatures when fused deposition modeling (FDM) is used, which could affect the bioactivity of the hydrogels. We will make effort to find suitable 3D-printing methods to mimic the structures of native cartilage for material design in the future. We have reorganized the discussion section based on the reviewer's suggestions.

8. Quite a few details on experimental design (e.g., treatment groups for in vivo studies) are relegated to the Supplementary section and should be moved to the main body of the manuscript.

Our reply: *Thanks for your comments. We have reorganized the experimental section and made revisions in the revised Manuscript and Supplementary section based on the reviewer's suggestions.*

Reviewer #2:

In the present work, the authors developed an ultra-durable cell-free bioactive hydrogel scaffold with fast shape memory and on-demand drug release for directional migration and differentiation of stem cells to promote cartilage regeneration. This hydrogel may be a new way to address the problem of cartilage regeneration. After carefully evaluating the manuscript, the manuscript in the present form will not be considered for publication in Nature Communications.

1. The experiments of antibacterial activity just investigated by spread plate method that was not enough. SEM and live/dead stain fluorescent test should be used to observe the morphology of the bacteria.

Our reply: *Thanks for your comments. SEM observation and Live/Dead staining were performed to further evaluate the anti-bacterial capability of the hydrogels. The results in Supplementary Fig. 15 showed that both E. coli and S. aureus lost their structural integrity after incubation with PTK hydrogel (10 mg/mL) and TA (5 mg/mL) for 12 h, and the bacterial structures remained intact in control and PMI hydrogel treated groups, suggesting the promising anti-bacterial efficiency by PTK hydrogel.*

Supplementary Fig. 15 SEM images of *E. coli* and *S. aureus* with different treatments.

Live/Dead staining results in Supplementary Fig. 16 revealed that KGN, PMI, and PKG hydrogels did not show antibacterial effects with strong green fluorescence (live bacteria). However, when the bacteria were treated with TA (5 mg/mL), PTA (10 mg/mL), and PTK (10 mg/mL) hydrogels, obvious red fluorescence (dead bacteria) was observed 12 h after incubation, which was attributed to the presence of TA in the hydrogel matrix (*Carbohyd Polym.* 2019, 224, 115147; *Carbohyd Polym.* 2022, 285, 119235; *ACS Appl Mater Interfaces.* 2016, 8(42), 28511-28521).

Supplementary Fig. 16 Live/Dead staining of *E. coli* and *S. aureus* with different treatments.

2. In vitro, the ability of PTK to recruit and promote differentiation of bone marrow mesenchymal stem cells to chondrocytes was verified.

Our reply: Thanks for your comments. To investigate the in vitro chondrogenic differentiation, the in vitro BMSC pellet culture test was performed as you suggested. The results in Supplementary Fig. 18 revealed that PTK hydrogel could induce larger cartilage pellet formation than that in the control group and PMI hydrogel-treated group, which indicated the promising chondrogenesis property of PTK hydrogel. We have made revision in the revised Manuscript and Supplementary Information.

Supplementary Fig. 18 *In vitro* chondrogenic differentiation mediated by PMI and PTK hydrogels.

3. All Figures in the manuscript are not clear enough, and the quality of the picture could be improved.

Our reply: *Thanks for your comments. We have improved the Figures' quality and the original Figures were also separately uploaded.*

4. Please confirm the accuracy of the sequential labeling of the pictures in the manuscript. For example, the antibacterial content in line 265 should correspond to Fig. 3g, rather than Fig. 2g.

Our reply: *Thanks for your comments. We have made revision in the revised Manuscript.*

5. The long-term cytotoxicity should be evaluated for gels, which demonstrates whether the degradation products have low toxicity.

Our reply: *Thanks for your comments. Since the long-term in vitro incubation of cells could cause cell death, we performed the in vivo biocompatibility of the PMI, PKG, PTA, and PTK hydrogels by subcutaneous implantation to investigate the toxicity to the main organs of rats. Two months after in vivo implantation of the hydrogels, the main organs of rats were collected and histologically evaluated. As shown in Supplementary Fig. 12, no obvious toxic reaction was identified in all groups, indicating the good biocompatibility of the hydrogels. We have made revision in the revised Manuscript and Supplementary Information.*

Supplementary Fig. 12 Long-term *in vivo* biocompatibility of PMI, PKG, PTA, and PTK hydrogels. (Scale bar: 200 μm)

6. How can hydrogels dissolve into solutions? How do you make the concentration gradient difference? Hydrogel is a gelatinous substance. Please explain why mg/mL is used to express the content of hydrogel.

Our reply: Thanks for your comments. As described in the section on supplementary methods, the pristine PMI polymer was synthesized by polyaddition reaction in DMF, and then KGN (40 mg) was directly added to the polymer solution (20 mL) followed by vacuum drying to obtain the drug-loaded PKG film. Therefore, 2 mg/mL represented the concentration of KGN in the PMI solution. Since KGN is hydrophobic, the swelling process should not induce severe release. Therefore, for the *in vivo* cartilage regeneration test, the implanted hydrogel (PTK hydrogel) contained approximately 0.008 mg of KGN. We have made clarification in the revised Supplementary Information.

7. Please explain how to ensure the rapid shape memory action and the feasibility of manipulation of the material in room temperature within 30 seconds. Does rapid thermo-triggered properties limit its practical application?

Our reply: Thanks for your comments. The shape memory process was recorded by a digital camera. As the reviewer mentioned, the clinical situation is complicated, and the fast shape memory process of PTK hydrogel may be not beneficial to the transfer process by the endoscope. To address this issue, further study should be focused on the following two aspects: 1. The shape recovery speed is dependent on temperature, and a higher temperature leads to faster shape recovery. Thus, the material design should aim to improve the hydrogen bonding interactions to elevate the transition temperature of the network restructuring, by which the scaffold should be stable during endoscopy transfer and recover to the initial shape in an appropriate duration in the injury site. 2. The design of heat insulation coating on the endoscope may be an alternative way to avoid undesired shape recovery during operation. We have added the discussion in the revised Manuscript.

8. Antibacterial experiments should add TA group and KGN group to prove which the component has antibacterial effect.

Our reply: Thanks for your comments. We have conducted the antibacterial studies of TA and KGN, and the results were shown below for your reference. It was found that the antibacterial efficiency reached 90% when the concentration of TA was 5 mg/mL, and KGN could not inhibit the proliferation of *E. coli* and *S. aureus* with the concentration of 100 mmol/L.

Fig. 3f Antibacterial test of TA, KGN, PMI, PKG, PTA, and PTK hydrogels.

Fig. 3g *Quantitative antibacterial efficiency of TA, KGN, PMI, PKG, PTA, and PTK hydrogels.*

9. In vitro stem cell experiments are needed to demonstrate whether the material induces cartilage tissue generation.

Our reply: *Thanks for your comments. As you suggested, we further evaluated the in vitro cartilage differentiation induced by PTK hydrogel through the BMSC pellet culture test. We have added these results in the revised Manuscript and Supplementary file (Supplementary Fig. 18). In the three-dimensional BMSC pellet culture (21 days), a larger cartilage pellet was observed in the PTK hydrogel treated group, which further illustrated the cartilage regeneration property of PTK hydrogel.*

Supplementary Fig. 18 *In vitro chondrogenesis evaluation mediated by PMI and PTK hydrogels.*

10. PET characterization method is novel, but what is the connection with PTK?

Our reply: *Thanks for your comments. Because of the simultaneous loading of TA and KGN, the PTK hydrogel possessed BMSCs homing and cartilage differentiation properties. In the surgical-induced cartilage defect rat model, the cartilage regeneration activity was significantly upregulated with the treatment of PTK hydrogel. The micro-PET evaluation is a novel method for bioactive evaluation, which could not only avoid the sacrifice of the experimental animal but also reveal the accurate bioactive intensity in living animals. Quantitative Na¹⁸F-based PET/CT has been reported to be useful for the assessment of metabolic, degenerative, traumatic, and neoplastic bone diseases (Semin Nucl Med. 2015. 45(1): 58–65). The standardized uptake value of osteochondral composites depends on the blood flow to the cartilage defect area, exposed bone surface area, regional osteochondral activity, and renal clearance (J Nucl Med. 2012. 53(8):1175-1184). The rapid focal uptake of Na¹⁸F occurs preferentially at sites with high osteo-chondrogenesis activity. Hence, the component of osteochondral turnover measured by ¹⁸F imaging is related to the chondrogenesis activity. Moreover, as the inflammation reaction was alleviated 4 weeks after surgery, the blood flow in different groups primarily reflected the cartilage regeneration activity. Therefore, the micro-PET evaluation could give a useful method*

to reflect the cartilage regeneration capability mediated by PTK hydrogel in living animals. We have added the background in the revised Manuscript.

11. The topic of the article has to do with on-demand drug release, but there is little research on this content in the in vitro drug release section. The encapsulation and loading rates of the drug were not clearly studied, and the cause of TA's sudden release in a short time was not explained.

Our reply: Thanks for your comments. The drug loading content of different hydrogels was recorded by measuring the weight change before and after drug loading. It was found that there was no obvious difference in TA loading content for PTA (36.1%) and PTK hydrogels (35.6%). Since KGN is hydrophobic, the further immersion in water and TA solution of PKG dry film did not induce the serious release of KGN, in which the drug loading content for PKG and PTK hydrogels was 0.1% and 0.1%, respectively.

a) TA loading content of PTA and PTK hydrogel. b) KGN loading content of PKG and PTK hydrogel.

The burst release of TA in a short time was attributed to the following aspects: 1. Some of TA is free in the hydrogel matrix and could be released first due to its high hydrophilic nature; 2. TA was also encapsulated in the hydrogel matrix by hydrogen bonding interaction, and the presence of different salts in the release buffer (PBS) could disrupt the physical interactions and lead to the fast release of TA in the surface layer. 3. Although TA could form multiple hydrogen bonds with the PMI, single or double hydrogen bonds were still able to be formed with relatively weak interaction, which could be easily broken to release free TA. The TA inside the hydrogel matrix with multiple hydrogen bonds with PMI was further released in a sustained manner. We have made revision in the revised Manuscript.

12. In the studies of anti-inflammatory, anti-oxidation and antibacterial effects of materials, the full manuscripts only emphasized the effect of the encapsulated drugs, but did not discuss the influence of the carrier on these aspects.

Our reply: *Thanks for your comments. As you suggested, we evaluated the anti-inflammatory, anti-oxidation, and antibacterial effects of PMI hydrogel. As shown in Fig 3e and 3f, the anti-inflammatory and antibacterial of PMI hydrogel were evaluated. These results revealed that the PMI hydrogel did not show obvious anti-inflammatory and antibacterial capability compared to the control group. The anti-oxidation property of the PMI hydrogel was also evaluated by the DPPH scavenging test (Fig 3d), and the DPPH scavenging efficiency was as low as 5.12 %, indicating poor anti-oxidation ability. We have added these results in the revised Manuscript (Fig 3d).*

Fig. 3d *Anti-oxidation property of PTK and PMI hydrogel.*

Reviewer #3:

The manuscript by Shuang Wang et al. is concerned with cell-free transplantable hydrogel scaffold for the cartilage regeneration by stem cell recruitment and its differentiation. In particular, the author presents the novel shape-memory drug-loaded hydrogel to facilitate cell migration and stem cell differentiation to chondrocyte. They employed tannic acid (TA) kartogenin (KGN) for multiple purposes to induce cartilage regeneration. They also show hydrogel-mediated cartilage regeneration using model organisms indicating the formation of neocartilage in vivo. This paper has great significance in that it presented hydrogen bond-based stage-dependent drug release from namely PTK hydrogel and consequently it induced homing of bone marrow stem cells followed by their differentiation; therefore this work would attract many researchers' attention to challenge current huddles to restore the defects on articular cartilages. However, there are a couple of points to be elucidated and confirmed. Suggestions regarding the manuscript are below:

1. PTK film was fabricated by drying after polycondensation of monomers followed by KGN and TA adsorption. This indicates the randomness of hydrogel structure in manufacturing. However there is a certain level of directionality in articular/hyaline cartilage structure, which is horizontally aligned chondrocytes in superficial zone and vertically aligned chondrocytes in middle/deep zone. Is it possible to 3D-print this hydrogel to render some level of aligned structure and to mimic structural, functional and mechanical characteristics of native cartilage tissue?

Our reply: *Thanks for your comments. 3D printing of this hydrogel to mimic the cartilage structure should be more promising for cartilage engineering. We found that the injection of PMI or PKG solution in DMF to TA solution could form hydrogels, but the dimension was not stable. In addition, TA may be oxidized at high temperatures when fused deposition modeling (FDM) is used, which could affect the bioactivity of the hydrogels. According to the reviewer's suggestions, we will make effort to find a suitable 3D-printing method to mimic the structures of native cartilage for material design in the future.*

2. The author explained the expected function of TA in PTK hydrogel. Even though the author listed a couple of references for the feature of TA to induce cell migration in different cell types, the author still needs to show additional references for TA to regulate inflammation and to reduce oxidative stress.

Our reply: *Thanks for your comments. In addition to the ELISA assay of IL-6 and DPPH scavenging test we performed in our work to verify the anti-inflammation and anti-oxidation of TA included hydrogels, recent publications that confirmed the bioactivities of TA were also added in the revised Manuscript (ACS Nano. 2020, 28, 14(7):8202-8219; Bioact Mater. 2021, 26, 9: 461-474; Acta Biomater. 2021, 126:119-131. Adv Healthc Mater. 2021, 10(3):e2001122).*

3. The author presented the antibacterial capability by testing the bacterial colony formation unit (CFU) counting method. If the bactericidal effect came from TA, the author should also show a PTA-treated group that consists of hydrogel only with TA to claim the antibacterial effect of TA.

Our reply: *Thanks for your comments. We have conducted the in vitro antibacterial experiments of PTA hydrogel and TA solution for your reference by different methods (spread plate, SEM observation, and Living/Dead staining). It was found that PTA hydrogel showed similar antibacterial efficiency to PTK hydrogel by spread plate method. And the same results for PTA hydrogel were also reported in our previous*

study (*Bioact. Mater.* 2021, 6, 3962–3975). We have added these results in the revised Manuscript (Fig. 3).

Fig. 3f Antibacterial test of TA, PMI, PKG, KGN, PTA, and PTK hydrogels.

Fig. 3g Quantitative antibacterial efficiency of TA, KGN, PMI, PKG, PTA, and PTK hydrogels.

SEM observation and Live/Dead staining were performed to further evaluate the anti-bacterial capability of the hydrogels. The results in Supplementary Fig. 15 showed that both *E. coli* and *S. aureus* lost their structural integrity after incubation with PTK hydrogel (10 mg/mL) and TA (5 mg/mL) for 12 h, and the bacterial structures remained intact in control and PMI hydrogel treated groups, suggesting the promising anti-bacterial efficiency by PTK hydrogel.

Supplementary Fig. 15 SEM images of *E. coli* and *S. aureus* with different treatments.

Live/Dead staining results in Supplementary Fig. 16 revealed that KGN, PMI, and PKG hydrogels did not show antibacterial effects with strong green fluorescence (live bacteria). However, when the bacteria were treated with TA (5 mg/mL), PTA (10 mg/mL), and PTK (10 mg/mL) hydrogels, obvious red fluorescence (dead bacteria) was observed 12 h after incubation, which was attributed the presence of TA in the hydrogel matrix. (*Carbohydr Polym.* 2019, 224, 115147; *Carbohydr Polym.* 2022, 285, 119235; *ACS Appl Mater Interfaces.* 2016, 8(42), 28511-28521).

Supplementary Fig. 16 Live/Dead staining of *E. coli* and *S. aureus* with different treatments.

4. What is the mechanism for TA burst release? If it is gradient-driven release from the hydrogel, the author would consider covalent linkage of TA with cleavable sequence to hydrogel monomer to control the release rate of TA from hydrogel after transplant.

Our reply: Thanks for your comments. As shown in Fig. 3a, the release profile of TA can be divided into two stages. In the first stage, the burst release of TA in a short time was observed. In the hydrogel matrix, some of TA was free and could be released first due to its highly hydrophilic nature. And TA was also encapsulated in the hydrogel matrix by hydrogen bonding interaction, and the salts in the release buffer (PBS) could break the physical interactions and accelerate the release of TA in the surface layer. In addition, TA inside the hydrogel network may form weak interactions with PMI (single or double hydrogen bonds), which could be released preferentially. In the second stage, TA that formed stable hydrogen bonds with PMI was further released in a sustained manner due to the gradual disruption of the multiple hydrogen bonds. Moreover, there are no active groups in the system that can react with TA to form covalent linkage since the polymer films were dried at high temperature (70 °C) for 48 h before TA loading.

5. The shape memory test conducted in this study is quite short. If this test was for the possibility of hydrogel transport through endoscope, the author should be able to control shape recovery rate to avoid clogging of hydrogel in the endoscope in the minimally invasive surgery.

Our reply: Thanks for your comments. We agree with the reviewer that the fast recovery process may induce clogging during practical applications. The hydrogen bonds mediated shape memory is dependent on temperature, and the higher temperature could lead to faster shape recovery. Therefore, our further work for material design will aim to improve the hydrogen bonding interactions to elevate the transition temperature of the network restructuring to control shape recovery rate, by which the scaffold should be stable during endoscopy transfer and recover to the initial shape in an appropriate duration in the injury site. We have added the discussion in the revised Manuscript.

6. The main driving force for the hydrogel to hold TA is the hydrogen bonds between polycondensated PMI and TA, therefore the release of TA from hydrogel can be affected by pH changes. Environmental pH is one of the crucial parameters that greatly affects enzyme activity and cellular biochemical reactions involving tissue repair and homeostasis. Author should conduct a TA release test in different ranges of pH to consider comparable conditions with articular cartilage defect microenvironment.

Our reply: Thanks for your comments. As you suggested, we evaluated the *in vitro* TA release from PTK hydrogel at different pHs (6.0, 6.8, 7.0, and 7.4). The results exhibited that the lower pH could reduce TA release and the higher pH could slightly increase TA release. TA was loaded in the hydrogels through hydrogen bonds, the release behavior should be affected by the concentration of salts, which may interfere with hydrogen bond formation. Since the concentration of salts in the buffer solution with different pHs was similar, these changes in pHs did not alter the TA releasing behaviors significantly.

In vitro TA release from PTK hydrogel at different pHs (6.0, 6.8, 7.0, and 7.4)

7. The hydrogel pore size and porosity measurement is important to characterize the BMSC homing effect of hydrogels. The author needs to measure the pore sizes and porosities of PMI, PKG, PTA and PTK. The author should also track the changes of pore size/porosity upon pH changes.

Our reply: Thanks for your comments. We tested the microstructures of PMI, PKG, PTA, and PTK hydrogels at different pHs (6.0, 6.8, 7.0, and 7.4), and the pore sizes were summarized below. It was found that there was no significant difference in the microstructures and pore size for all the hydrogels at different pHs.

a) SEM images of PTK hydrogel at different pHs (6.0, 6.8, 7.0, and 7.4). b) Average pore size of PTK hydrogel at different pHs.

Reviewer #4:

Note: This review is limited to aspects of the manuscript pertaining to the use of PET imaging.

Summary: This manuscript presents the development of a hydrogel scaffold to promote cartilage regeneration. As part of the study, the authors used Na¹⁸F PET imaging to image bone remodeling non-invasively in mouse models. NaF is a well-established tracer, used to measure bone remodeling in vivo, including bone metastases and musculoskeletal disorders. The use of Na¹⁸F to image cartilage regeneration seems logical and is novel. Specific comments are provided below:

1. It would be helpful to provide additional background and rationale pertaining to the mechanism of deposition of Na¹⁸F during bone remodeling. ¹⁸F ion is chemically analogous to calcium and is incorporated into the bone matrix as fluoroapatite.

Our reply: Thanks for your comments. Na^{18}F is a bone-specific tracer first reported in 1962, and the tracer uptake by the osteochondral composites is due to chemisorption with the exchange of $^{18}\text{F}^-$ for OH^- on the surface of the hydroxyapatite osteochondral matrix. Quantitative Na^{18}F -based PET/CT has been reported to be useful for the assessment of metabolic, degenerative, traumatic, and neoplastic bone diseases (Semin Nucl Med. 2015. 45(1): 58–65). The standardized uptake value (SUV) of osteochondral composites depends on the blood flow to the cartilage defect area, exposed bone surface area, regional osteochondral activity, and renal clearance (J Nucl Med. 2012. 53(8):1175-1184.) The rapid focal uptake of Na^{18}F occurs preferentially at sites of high osteo-chondrogenesis activity where cartilage remodeling is greatest. Hence, the component of osteochondral turnover being measured by ^{18}F imaging is chondrogenesis activity. We have added the background in the revised Manuscript.

2. The following statement is incorrect: " ^{18}F -sodium fluoride (Na^{18}F) is the most commonly used PET tracer". The most common PET tracer is ^{18}F -fluorodeoxyglucose, which is used to image glucose metabolism.

Our reply: Thanks for your comments. As you suggested, we have made revision in the revised Manuscript.

3. The manuscript refers to Na^{18}F uptake as a marker of metabolic activity, but this may not be exact since ^{18}F ions bind to bone by chemisorption, therefore the uptake of the tracer depends primarily on the area of exposed bone surface.

Our reply: Thanks for your comments. Previous studies revealed that the rate-limiting step of Na^{18}F uptake is blood flow, and the initial Na^{18}F distribution represents blood flow that varies among different bones (J Nucl Med. 2010. 51(12):1826-9.) Therefore, the Na^{18}F uptake was chosen as a metabolic marker for cartilage regeneration evaluation in this study.

4. The details of the imaging experiments are vague, and the following information should be provided:

- Whether CT images were also acquired.

Our reply: Thanks for your comments. The CT evaluation was not included in this study because cartilage defects could not be clearly identified on X-ray-based tests.

5. Duration of each acquisition.

Our reply: Thanks for your comments. The acquisition duration was 30 minutes and the blood flow image was reconstructed using PMOD software.

6. Where the tracer was obtained / produced.

Our reply: Thanks for your comments. ^{18}F -fluoride is produced by 11-MeV proton irradiation of ^{18}O -water in a tantalum target body using a cyclotron. The irradiated aqueous solution containing ^{18}F -fluoride is diluted with sterile water (5 mL) and passed through a cation exchange (H^+ form) cartridge. The eluent from the cation exchange cartridge is passed through an anion exchange (HCO_3^- form) cartridge to trap the ^{18}F -fluoride. The anion exchange cartridge was flushed with 10 mL of sterile water, and the ^{18}F fluoride was then eluted with 10 mL of sterile normal saline and passed through a sterile filter into a sterile multidose vial for PET assay.

7. Image reconstruction and any filtering applied.

Our reply: Thanks for your comments. The image was reconstructed by three-dimensional order subset expectation maximization (3D-OSEM) using PMOD software.

8. How the regions of interest were drawn, and whether the analysis was performed on a 2D slice or on the full 3D volume.

Our reply: Thanks for your comments. The ROI was manually drawn, and the analysis was performed on the 3D volume. The standardized uptake value (SUV) of the cartilage tissues was further evaluated and shown below to exclude potential bias caused by ROI drawing. SUV was calculated by the following formula:

$$SUV = \frac{\text{region of interest activity concentration (MBq/mL)}}{\text{injection activity concentration } \left(\frac{\text{MBq}}{\text{mL}}\right) / \text{body weight (kg)}}$$

SUV value of control, PMI, PTA, PKG, and PTK hydrogel treated groups.

9. Whether decay correction was applied.

Our reply: *Thanks for your comments. The decay correction was not applied in this study because all the PET acquisition was performed within 30 minutes and the decay rate was negligible.*

10. There seems to be a discordance between the images and the quantitative analysis. The analysis suggests a 9 fold difference between PTK hydrogel and control, however, the images do not show an increase of signal of that magnitude. At most there is maybe a 2x enhancement of the signal (based on the color scale). Furthermore, the increase is unlikely to be specific to the treatment since an increase is also observed in the untreated contralateral knee.

Our reply: *Thanks for your comments. The color scale only reflects the relative uptake difference in each group. The quantitative analysis of nuclide accumulation was calculated by software integration while the relative nuclide accumulation in PTK treated group exceeded the range of the color bar which is 8.8 times higher than that in the control group. The increase observed in the untreated contralateral knee was due to individual basic metabolism differences which were corrected by relative nuclide accumulation.*

11. Could the ROI analysis be applied to the contralateral knee as a control?

Our reply: *Thanks for your comments. The ROI was a commonly used method to set measurement threshold which could also be applied to the control group (J Clin Med Res. 2014. 6(6): 435-42.).*

12. The rainbow color scale should include quantitative units for reference.

Our reply: *Thanks for your comments. As you suggested, we have added quantitative units to the rainbow color scale in the revised Manuscript.*

13. It is not clear if the images shown in b-c are maximum intensity projections or cross-sectional images. Also why is there no signal from the bladder? $^{18}\text{F-NaF}$ should have rapid renal clearance. Could additional orthogonal views be included to clarify where the area of interest (dated box) is located with respect to the body?

Our reply: *Thanks for your comments. The images in Fig 5 b-c were maximum intensity projections (MIP). The bladder was evacuated by external stimulation of the rat belly before PET evaluation to avoid potential bias. The knee area was stretched away from the rat body to avoid image interference and the renal area was not included in the Image acquisition area. The cross section of PTK group was shown below for your reference.*

The transverse, sagittal, and coronal plane of micro-PET acquisition.

14. The text should clarify how the "relative nuclide accumulation" was computed. Was the reference taken as the contralateral (healthy) knee from the same animals, or the damaged but untreated knee from the control animals? An internal reference should be used to normalize PET signal within each animal to account for possible variations in injected dose, etc.

Our reply: *Thanks for your comments. The relative nuclide accumulation was calculated by the following equations:*

$$\text{Relative nuclide accumulation} = \frac{Ae \div A(\text{spine})}{Ac \div A(\text{spine})} = Ae/Ac$$

The average nuclide accumulation of ROI in the experimental groups was marked as Ae, and the average nuclide accumulation of ROI in the control group was marked as Ac. The damaged but untreated knee was used as control. The nuclide uptake in the spine area was used for internal reference, which was applied to both the treated and untreated sides of each group. Therefore, the internal reference made no difference to the relative nuclide accumulation comparison.

15. The 30 min superposition may not be the best way to analyze the dynamic data. Kinetic analysis using a compartment model would be more accurate and could help distinguish between tracer delivery effects (blood flow) and tracer deposition into bone matrix. If this is not possible, the image obtained at 30 min would be more relevant since the earlier images will mainly show blood perfusion.

Our reply: *Thanks for your comments. Kinetic analysis was used in the first 20 min for blood flow evaluation as shown in Fig 5b, which could reflect the higher metabolic activity achieved on PTK hydrogel-treated side. However, because of the rapid renal clearance, the kinetic image for 30 minutes of the keen tissue was too dim to be further evaluated. Therefore, the superposition was used for general exhibitions.*

16. Minor points:

L348 reflex  reflect

L369: ... dash  dashed line

Our reply: *Thanks for your comments. We have made revision in the revised Manuscript.*

REVIEWER COMMENTS

Reviewer #2 (Remarks to the Author):

The authors have answered my question, I have no further question. The revised manuscript in the present form can be published on this journal.

Reviewer #3 (Remarks to the Author):

The authors address all the issues that this reviewer suggested. Therefore, the manuscript is ready to be published.

Reviewer #4 (Remarks to the Author):

The manuscript has been substantially revised their manuscript, and many of the previous points have been addressed satisfactorily. However, I would like to go back to a few comments from the previous review.

In Figure 5d, it is challenging to accept that the uptake increases 9 fold (i.e. 800%) between the control group and the PTK treated group. This was a very large increase, and in my experience, a 9x increase in uptake should result in a more visible difference on the PET images.

In the response, it is mentioned that the the color scale only reflects the relative uptake difference in each group and that the nuclide accumulation in PTK treated group exceeded the range of the color bar. Best practice for PET imaging is, when possible, to use absolute units (kBq/cc or SUV) and display all images on the same intensity scale, so that a direct comparison of the uptake is possible. If the intensity exceeds the range of the color bar, then the intensity scale should be adjusted to avoid saturation of the signal of interest.

In addition, an internal reference should taken within each image to normalize any differences between acquisitions. The authors used the uptake in the healthy spine as their reference, which is suitable here. However, the "relative nuclide accumulation" (defined in the Method section) removed the internal reference from the calculation and only considered the ratio A_e / A_c . The spine internal reference cannot be removed from the equation because, in general, $A_e(\text{spine}) \neq A_c(\text{spine})$ therefore the fraction cannot be simplified.

In the PET images, the spine uptake appears lower in the control group compared to the PTK group, which suggest that either injected dose was lower, the imaging timepoints was different, or the intensity display window settings are substantially different. This is also evidenced in the finding that the the (untreated, healthy) contralateral knee in the PTK-treated mice appears brighter than the injured knee in control mice. Additionally, the difference between injured and healthy knee in the PTK-treated mouse does not seem to support the claim of 9x increase in the treated knee.

To put the issue to rest, I suggest the following plan:

- 1- Replot the PET images using an absolute color scale (instead of a relative one), adjusting the scale to avoid saturation of the high intensity signal. The same display settings should be used for all images to emphasize the absolute uptake difference between the different group. Additionally, the color scale should include the unit used (e.g. kBq/cc, $\mu\text{Ci/mL}$, SUV, etc). Numbers without units cannot be interpreted.
- 2- In the calculation of the relative nuclide accumulation, the internal spine reference should be

included in the calculation. The formula in the Methods section must be corrected by removing " $= A_e/A_c$ " at the end of the equation. The same approach should also be used for the maximum uptake value.

3- If the authors stand by the claim that the uptake is increased 9-fold, then they should provide additional justification (in supporting information). The following information could be provided for each group:

- mean injected ^{18}F -NaF tracer dose per mouse (mean and standard deviation)
- mean uptake in the left injured knee ROI (in kBq/cc or similar unit of radiotracer concentration; mean and standard deviation)
- mean uptake in the right healthy contralateral knee ROI (in kBq/cc or similar unit of radiotracer concentration; mean and standard deviation)
- mean uptake in the spine ROI (in kBq/cc or similar unit of radiotracer concentration; mean and standard deviation)
- figure showing the ROIs used for one of the mice

Reviewer #5 (Remarks to the Author):

Below is an assessment of the authors' response to Reviewer 1's concerns.

Comment 1. The grammar in the manuscript is improved.

Comment 2. The reviewer requests clarification in the methods regarding the type and concentration of hydrogels used in the in vivo studies.

These methods appear to be located in the supplemental methods. While this non-expert would not be able to repeat such fabrications, someone familiar in the art, likely thinks this is sufficient.

Comment 3. The authors provide the requested additional experimental data regarding the mechanical properties of the biomaterials.

Comment 4. The authors provide information requested regarding weight bearing of the knee joints.

Comment 5

'The authors state that the higher levels of metabolism measured by PET is due to the chondrogenic activity of the cells and increased blood flow. Since this is a fairly new, non-standard way of assessing cartilage regeneration, a more thorough understanding of how the various cells in the joint may be contributing to metabolic activity is required.'

The authors response states that "The standardized uptake value of osteochondral composites depends on the blood flow to the cartilage defect area, exposed bone surface area, regional osteochondral activity, and renal clearance (J Nucl Med. 2012. 53(8):1175-1184.). The rapid focal uptake of ^{18}F Na occurs preferentially at sites with high osteo-chondrogenesis activity.", but perhaps incorrectly assumes this is due to chondrogenesis activity.

Also, ", the blood flow in different groups primarily reflected the cartilage regeneration activity" : Blood flow is also not a direct link to cartilage regeneration.

The response of the authors moreso describes the methodology rather than answering the reviewer's question. I believe the reviewer was requesting information about the functions of cells that lead to increases in metabolic activity quantified by uptake of the dye, and whether this function is differentiation of progenitor cells (presumably MSCs) into metabolically active chondrocytes. The authors infer chondrogenesis without demonstrating cell function. I think replacing 'chondrogenesis' with 'bioactivity' is appropriate in these and other relevant instances.

Comment 6. MSC-based cartilage formation may be fibrocartilaginous or the cells may exhibit markers indicating maturation down the endochondral ossification pathway. Therefore, non-cartilaginous markers should be investigated with immunoblotting and staining.

The authors responded by providing staining for ALP, a bone marker, and Safranin O, which stains proteoglycans abundant in articular cartilage. I do not observe any positive ALP stain in the subchondral bone in Sup Fig 22, suggesting that their IHC for this protein did not detect the antigen in the positive control tissue, thus the concluded lack of positive staining in the neocartilage is premature. I believe the reviewer was asking for something like COL1 staining for fibrocartilage, and COLX staining for hypertrophy/endochondral initiations. A high magnification saf O stain would provide morphological information to the reviewer that indicates whether hypertrophy is occurring, but fibrocartilage requires a type I collagen stain.

Comment 7. Some discussion points identified by this reviewer remain in the discussion. Highlighted example page 8. Second example page 13.

Reviewer #2:

The authors have answered my question, I have no further question. The revised manuscript in the present form can be published on this journal.

Our reply: Thanks very much.

Reviewer #3:

The authors address all the issues that this reviewer suggested. Therefore, the manuscript is ready to be published.

Our reply: Thanks very much.

Reviewer #4:

The manuscript has been substantially revised their manuscript, and many of the previous points have been addressed satisfactorily. However, I would like to go back to a few comments from the previous review.

In Figure 5d, it is challenging to accept that the uptake increases 9 fold (i.e. 800%) between the control group and the PTK treated group. This was a very large increase, and in my experience, a 9x increase in uptake should result in a more visible difference on the PET images.

In the response, it is mentioned that the color scale only reflects the relative uptake difference in each group and that the nuclide accumulation in PTK treated group exceeded the range of the color bar. Best practice for PET imaging is, when possible, to use absolute units (kBq/cc or SUV) and display all images on the same intensity scale, so that a direct comparison of the uptake is possible. If the intensity exceeds the range of the color bar, then the intensity scale should be adjusted to avoid saturation of the signal of interest.

In addition, an internal reference should taken within each image to normalize any differences between acquisitions. The authors used the uptake in the healthy spine as their reference, which is suitable here. However, the "relative nuclide accumulation" (defined in the Method section) removed the internal reference from the calculation and only considered the ratio A_e / A_c . The spine internal reference cannot be removed from the equation because, in general, $A_e(\text{spine}) \neq A_c(\text{spine})$ therefore the fraction cannot be simplified.

In the PET images, the spine uptake appears lower in the control group compared to the PTK group, which suggest that either injected dose was lower, the imaging timepoints was different, or the intensity display window settings are substantially different. This is also evidenced in the finding that the the (untreated, healthy) contralateral knee in the PTK-treated mice appears brighter than the injured knee in control mice. Additionally, the difference between injured and healthy knee in the PTK-treated mouse does not seem to support the claim of 9x increase in the treated knee.

To put the issue to rest, I suggest the following plan:

1. Replot the PET images using an absolute color scale (instead of a relative one), adjusting the scale to avoid saturation of the high intensity signal. The same display settings should be used for all images to emphasize the absolute uptake difference

between the different group. Additionally, the color scale should include the unit used (e.g. kBq/cc, $\mu\text{Ci}/\text{mL}$, SUV, etc). Numbers without units cannot be interpreted.

Our reply: Thanks for your comments. As you suggested, we have unified the display settings for all treatment groups and used the absolute uptake for the color scale (as shown below). The average absolute uptake value in the PTK group is 1498.13 kBq/cc while the average absolute uptake value in PTK treated group spine is 206.19 kBq/cc. Meanwhile, the average absolute uptake value in the control group is 209.97 kBq/cc while the average absolute uptake value in control group spine is 203.06 kBq/cc. Therefore, the Relative nuclide accumulation value of PTK is 7.03 based on the following equations:

$$\text{Relative nuclide accumulation} = \frac{Ae \div Ae(\text{spine})}{Ac \div Ac(\text{spine})}$$

Fig. 5c Nuclide accumulation in control, PMI, PKG, PTA, and PTK groups. As you suggested, we have made revisions in the revised Manuscript.

2. In the calculation of the relative nuclide accumulation, the internal spine reference should be included in the calculation. The formula in the Methods section must be corrected by removing " $= Ae/Ac$ " at the end of the equation. The same approach should also be used for the maximum uptake value.

Our reply: Thanks for your comments. As you suggested, we have made revision in the revised Manuscript and recalculated relative nuclide accumulation and maximum uptake values as shown below and in Fig. 5d and 5e.

Fig. 5d Relative nuclide accumulation values in control, PMI, PKG, PTA, and PTK groups. **Fig. 5e** Maximum uptake values in control, PMI, PKG, PTA, and PTK groups.

3. If the authors stand by the claim that the uptake is increased 9-fold, then they should provide additional justification (in supporting information). The following information could be provided for each group:

- mean injected ¹⁸F-NaF tracer dose per mouse (mean and standard deviation)
- mean uptake in the left injured knee ROI (in kBq/cc or similar unit of radiotracer concentration; mean and standard deviation)
- mean uptake in the right healthy contralateral knee ROI (in kBq/cc or similar unit of radiotracer concentration; mean and standard deviation)
- mean uptake in the spine ROI (in kBq/cc or similar unit of radiotracer concentration; mean and standard deviation)
- figure showing the ROIs used for one of the mice.

Our reply: *Thanks for your comments. The required data was shown below. The recalculated relative nuclide accumulation values were calculated accordingly.*

Supplementary Table S5 Summary of the results for *in vivo* Micro PET evaluation.

	mean injected ¹⁸ F-NaF tracer dose (uCi)	standard deviation	left injured knee ROI (kBq/cc)	standard deviation	contralateral knee ROI (kBq/cc)	standard deviation	spine (kBq/cc)	standard deviation
Control	1203	107	209.96	12.15	201.05	11.1	203.06	10.05
PMI	1125	114	283.28	11.95	215.33	18.46	251.34	12.23
PKG	1230	110	561.81	20.38	221.59	18.62	216.47	18.06
PTA	1307	105	1282.53	87.62	251.32	21.08	233.15	15.41
PTK	1258	113	1498.13	85.33	218.14	19.15	206.19	12.77

The ROI was manually drawn as shown below, and the same acquisition area size was used in all experiment groups.

The ROI used in Micro PET evaluation

Fig. 5 Macroscopic observation and radionuclide image of in vivo metabolic activity. *a* Gross observation of the rat articular specimens at 0 and 8 weeks after operation with different treatments. Scale bar: 2 mm. *b* Dynamic Na^{18}F uptake in the defect at rats with the treatment of PTK hydrogel and the superposition of images 30 min after injection. *c* The accumulation of Na^{18}F in the injury sites 30 minutes after injection. Scale bar: 5 mm. *d* Quantitative analysis of the nuclide accumulation of ROI. *e* Quantitative analysis of the maximum nuclide accumulation of ROI (Relative and maximum Na^{18}F uptake in ROI was automatically segmented by PMOD). Data are presented as the mean \pm SD ($n = 5$. * $P < 0.05$, ** $P < 0.01$, *** $P < 0.001$).

Reviewer #5:

Comment 1. The grammar in the manuscript is improved.

Our reply: Thanks very much.

Comment 2. The reviewer requests clarification in the methods regarding the type and concentration of hydrogels used in the in vivo studies.

These methods appear to be located in the supplemental methods. While this non-expert would not be able to repeat such fabrications, someone familiar in the art, likely thinks this is sufficient.

Our reply: Thanks very much.

Comment 3. The authors provide the requested additional experimental data regarding the mechanical properties of the biomaterials.

Our reply: Thanks very much.

Comment 4. The authors provide information requested regarding weight bearing of the knee joints.

Our reply: Thanks very much.

Comment 5

‘The authors state that the higher levels of metabolism measured by PET is due to the chondrogenic activity of the cells and increased blood flow. Since this is a fairly new, non-standard way of assessing cartilage regeneration, a more thorough understanding of how the various cells in the joint may be contributing to metabolic activity is required.’

The authors’ response states that “The standardized uptake value of osteochondral composites depends on the blood flow to the cartilage defect area, exposed bone surface area, regional osteochondral activity, and renal clearance (J Nucl Med. 2012. 53(8):1175-1184.). The rapid focal uptake of Na¹⁸F occurs preferentially at sites with high osteo-chondrogenesis activity.”, but perhaps incorrectly assumes this is due to chondrogenesis activity.

Also, “, the blood flow in different groups primarily reflected the cartilage regeneration activity” : Blood flow is also not a direct link to cartilage regeneration. The response of the authors moreso describes the methodology rather than answering the reviewer’s question. I believe the reviewer was requesting information about the functions of cells that lead to increases in metabolic activity quantified by uptake of the dye, and whether this function is differentiation of progenitor cells (presumably MSCs) into metabolically active chondrocytes. The authors infer chondrogenesis without demonstrating cell function. I think replacing ‘chondrogenesis’ with ‘bioactivity’ is appropriate in these and other relevant instances.

Our reply: Thanks for your comments. The blood flow difference is a common approach to evaluate the metabolic activities in specific tissues. As the reviewer mentioned, the metabolic activity is more complicated rather than inflammation and cartilage

regeneration, which could be affected by various bioactivities such as subchondral bone remodeling, local edema, post-surgery activities et al. Therefore, the micro-PET evaluation could not completely distinguish the difference between these bioactivities when applied to living animals. We agreed with the reviewer that the micro-PET results could not direct to “chondrogenesis” and we have replaced it with “bioactivity” as the reviewer recommended.

Comment 6. MSC-based cartilage formation may be fibrocartilaginous or the cells may exhibit markers indicating maturation down the endochondral ossification pathway. Therefore, non-cartilaginous markers should be investigated with immunoblotting and staining.

The authors responded by providing staining for ALP, a bone marker, and Safranin O, which stains proteoglycans abundant in articular cartilage. I do not observe any positive ALP stain in the subchondral bone in Sup Fig 22, suggesting that their IHC for this protein did not detect the antigen in the positive control tissue, thus the concluded lack of positive staining in the neocartilage is premature. I believe the reviewer was asking for something like COL1 staining for fibrocartilage, and COLX staining for hypertrophy/endochondral initiations. A high magnification saf O stain would provide morphological information to the reviewer that indicates whether hypertrophy is occurring, but fibrocartilage requires a type I collagen stain.

Our reply: Thanks for your comments. As the reviewer suggested, we performed ELISA test to investigate the expression of COL I in different groups. As shown below, there was no obvious difference in the expression level of COL I by ELISA test for the untreated healthy cartilage and PTK hydrogel-induced new cartilage tissue.

Supplementary Fig. 22e ELISA test of COL I expression in untreated healthy cartilage and PTK hydrogel treated groups.

Comment 7. Some discussion points identified by this reviewer remain in the discussion. Highlighted example page 8. Second example page 13.

Our reply: Thanks for your comments. We have made revisions in the revised Manuscript following the suggestion of **Reviewer #1**, moved the tissue adhesiveness and shape memory properties descriptions to the discussion section, removed main organ adhesive photographs, and reorganized our description of the MSC migration on page 8. On page 13, we have moved the description of over-expressed ROS in the cartilage defect area to the discussion section as the reviewer suggested.

REVIEWER COMMENTS

Reviewer #4 (Remarks to the Author):

I appreciate the effort by the authors to clarify their analysis of the PET images. However, I am still not convinced that the data supports the claim of a 7-fold increase in [¹⁸F]NaF in the regenerating cartilage using bioactive hydrogel scaffold. [¹⁸F]NaF has high baseline uptake in normal bone, and an increase of this magnitude would be unexpected.

The issue of the high uptake in the contralateral knee, which was obvious in the previous version of the article, was not addressed directly. If the PET signal indicates cartilage regeneration, how do the authors explain the similarly high signal in the (uninjured and untreated) contralateral knee?

Instead, the PET images in Fig. 5c have been changed where only a narrow slice of the mouse volume is now displayed, which no longer includes the contralateral knee and most of the bony anatomy. Since the signal from the contralateral knee was the primary issue, the new images are unhelpful and in fact conceal the magnitude of the problem. The comparison between Fig 5b and 5c shows that in the first sets images, the entire bony anatomy is visible whereas in the second set of images, only a very narrow section of the spine can be seen.

As I remain skeptical of the treatment and analysis of the images despite the efforts of the authors, I believe the only way to publish this paper is by releasing the raw 3D PET images (DICOM format; at least one mouse per group) via a scientific repository. Readers of the article can then check the raw data and make their own conclusion.

Reviewer #5 (Remarks to the Author):

The manuscript was improved as requested. I have no additional comments or requests for the authors.

Reviewer #4

I appreciate the effort by the authors to clarify their analysis of the PET images. However, I am still not convinced that the data supports the claim of a 7-fold increase in [^{18}F]NaF in the regenerating cartilage using bioactive hydrogel scaffold. [^{18}F]NaF has high baseline uptake in normal bone, and an increase of this magnitude would be unexpected.

The issue of the high uptake in the contralateral knee, which was obvious in the previous version of the article, was not addressed directly. If the PET signal indicates cartilage regeneration, how do the authors explain the similarly high signal in the (uninjured and untreated) contralateral knee?

Instead, the PET images in Fig. 5c have been changed where only a narrow slice of the mouse volume is now displayed, which no longer includes the contralateral knee and most of the bony anatomy. Since the signal from the contralateral knee was the primary issue, the new images are unhelpful and in fact conceal the magnitude of the problem. The comparison between Fig 5b and 5c shows that in the first sets images, the entire bony anatomy is visible whereas in the second set of images, only a very narrow section of the spine can be seen.

As I remain skeptical of the treatment and analysis of the images despite the efforts of the authors, I believe the only way to publish this paper is by releasing the raw 3D PET images (DICOM format; at least one mouse per group) via a scientific repository. Readers of the article can then check the raw data and make their own conclusion.

Our reply: *Thanks for your comments. Na^{18}F uptake could increase drastically in bioactivities involving blood flowing increasing, such as tumor, inflammation, and tissue healing. The uptake values could be elevated by more than 10 times in some cases (Clin Radiol. 2022. 77: 613-620). The RAW data was presented in MPV in our equipment and all RAW data were uploaded in Source Data. The full acquisition steps were shown below for your reference.*

Acquisition steps of micro-PET.

Reviewer #5:

The manuscript was improved as requested. I have no additional comments or requests for the authors.

Our reply: *Thanks very much.*

REVIEWER COMMENTS

Reviewer #4 (Remarks to the Author):

Ref: NCOMMS-22-44392C

In response to my previous comments, the authors have shared raw PET image data (in "MPV" format) from their study. The addition of these data is helpful to better understand the effect of the treatment on knee regeneration as assessed by PET imaging.

However, importing data using raw vendor-specific format is not without effort and is unlikely to be helpful for most readers unless further information is provided. I disagree with the statement that "The MPV format we used is also well-accepted for image process in clinic and research." The MPV format is used by one very small vendor of pre-clinical PET scanner (Shandong MadiC Technology Co) and it is not used clinically anywhere. The DICOM format is the standard format for sharing medical image. Unlike MPV, DICOM contains the metadata required to properly interpret the images, including pixel grid size and calibration factors. The MPV format only contains the raw image matrix data without the associated information. I understand that the authors may have encountered some challenge converting the MPV data into DICOM. The PMOD software, which they used, should be able to convert the full volume data into DICOM format. The DICOM provided by the authors were screen shots of the final analysis, not the full 3D volume data.

After some effort, it was possible to load MPV data for analysis although the calibration and scaling factors were not present in the MPV file, which prevents absolute quantitation of the PET signal.

The first finding, which I previously missed, is that the mice are shown upside-down in the manuscript. For the sake of clarity, it would be helpful to rotate all PET images so that the tail and feet are at the bottom of the image, and the spine and knees at the top (see slide 2 in attachment). This is a minor point.

The second issue is that the MPV filenames do not match the labels in Figure 5c (see slides 3-7 in attachment). For instance, the file named CONTROL.MPV actually shows the image labeled "PMI" in the manuscript. This issue is perplexing, and it is possible that the MPV files were incorrectly named or the figures were incorrectly labeled.

Third, region-of-interest (ROI) analysis of the MPV data shows there is no significant difference in PET signal between treated and untreated knee (see slides 3-7 in attachment). The variation between the two sides is on the order of 10-20 %, which is far below the 7-fold enhancement that was claimed in the manuscript. This difference is likely within normal variability. There is also no visible difference in signal between the two knees. On the basis of these images, it is not possible to conclude that the treatment led to a significant enhancement in the PET signal. Both knees have high PET signal so a massive increase in signal as previously reported is not likely. While I can only guess, possibly the issue was ROI analysis was performed on a single slide, with the contralateral knee outside the area being inspected.

This issue was raised multiple times throughout the review process but was not addressed by the authors. The raw data provided by the authors do not support the claim of a several-fold increase in PET signal in the treated knee. The authors also did not explain why, in Fig 5c, they changed the slice thickness between the original submission and the second revision, which obscured the fact that both knees had similarly strong PET signal. Unfortunately, in view of these problems, I cannot endorse publication of this manuscript.

Reviewer #4:

In response to my previous comments, the authors have shared raw PET image data (in “MPV” format) from their study. The addition of these data is helpful to better understand the effect of the treatment on knee regeneration as assessed by PET imaging.

Our reply: *Thanks for your comments.*

However, importing data using raw vendor-specific format is not without effort and is unlikely to be helpful for most readers unless further information is provided. I disagree with the statement that “The MPV format we used is also well-accepted for image process in clinic and research.” The MPV format is used by one very small vendor of pre-clinical PET scanner (Shandong Madic Technology Co) and it is not used clinically anywhere. The DICOM format is the standard format for sharing medical image. Unlike MPV, DICOM contains the metadata required to properly interpret the images, including pixel grid size and calibration factors. The MPV format only contains the raw image matrix data without the associated information. I understand that the authors may have encountered some challenge converting the MPV data into DICOM. The PMOD software, which they used, should be able to convert the full volume data into DICOM format. The DICOM provided by the authors were screen shots of the final analysis, not the full 3D volume data.

After some effort, it was possible to load MPV data for analysis although the calibration and scaling factors were not present in the MPV file, which prevents absolute quantitation of the PET signal.

Our reply: *Thanks for your comments. We agree with the reviewer that DICOM is the most well-accepted format for medical images. When we performed this study, the prototype of the equipment we used did not provide the DICOM output option. Based on your comment, we tried to convert the MPV format into DICOM, however, it caused inevitable data change and information loss (not the full 3D volume data). With the further update of our equipment and software platform in the future, we will try our best to collect data in a more universally accepted data form.*

The first finding, which I previously missed, is that the mice are shown upside-down in the manuscript. For the sake of clarity, it would be helpful to rotate all PET images so that the tail and feet are at the bottom of the image, and the spine and knees at the top (see slide 2 in attachment). This is a minor point.

Our reply: *Thanks for your comments. The exhibition position of the rats we chose was the surgery position which was coordinated with the macroscopic view of rat knees. The upright position view is shown below for your reference.*

The second issue is that the MPV filenames do not match the labels in Figure 5c (see slides 3-7 in attachment). For instance, the file named CONTROL.MPV actually shows the image labeled “PMI” in the manuscript. This issue is perplexing, and it is possible that the MPV files were incorrectly named or the figures were incorrectly labeled.

Our reply: *Thanks for your comments. We are sorry for the mistakes. We have made corrections for the mislabeled filenames and updated the original data.*

Third, region-of-interest (ROI) analysis of the MPV data shows there is no significant difference in PET signal between treated and untreated knee (see slides 3-7 in attachment). The variation between the two sides is on the order of 10-20 %, which is far below the 7-fold enhancement that was claimed in the manuscript. This difference is likely within normal variability. There is also no visible difference in signal between the two knees. On the basis of these images, it is not possible to conclude that the treatment led to a significant enhancement in the PET signal. Both knees have high PET signal so a massive increase in signal as previously reported is not likely. While I can only guess, possibly the issue was ROI analysis was performed on a single slide, with the contralateral knee outside the area being inspected.

Our reply: *Thanks for your comments. The selection of ROI area boundary could significantly affect the outcome of imaging analysis. Compared to the ROI the review used in the attachment (including the surrounding areas), in our research, the surgical-induced cartilage defect was relatively small (excluding the surrounding areas). We agree with the reviewer that the ROI labeled in slides 3-7 in the attachment should work for the analysis of relative nuclide accumulation, but may reduce the significance of the values because the ROI included more surrounding tissues. Moreover, in order to exhibit the nuclide uptake in the surgical sites, the unified quantitative display settings were applied in different groups which could dim the low uptake areas such as the spine or bony anatomy. In order to precisely make the conclusion for Fig 5c and 5d, we changed “It was found that the relative Na^{18}F accumulation in the PTK hydrogel-treated group was 7.0 and 6.4 times higher than that in the control group and PMI hydrogel-treated group, respectively” to “It was found that the relative Na^{18}F*

accumulation in the PTK hydrogel-treated group was higher than that in the control group and PMI hydrogel-treated group”.

This issue was raised multiple times throughout the review process but was not addressed by the authors. The raw data provided by the authors do not support the claim of a several-fold increase in PET signal in the treated knee. The authors also did not explain why, in Fig 5c, they changed the slice thickness between the original submission and the second revision, which obscured the fact that both knees had similarly strong PET signal. Unfortunately, in view of these problems, I cannot endorse publication of this manuscript.

Our reply: *Thanks for your comments. The reason we made changes in Fig 5c is that we wanted to highlight the uptake in the knee areas compared to the spine and bony anatomy areas. We need to claim that all the changes for Fig 5c were based on the unified quantitative display settings in different groups. We also changed the conclusion of “It was found that the relative Na¹⁸F accumulation in the PTK hydrogel-treated group was 7.0 and 6.4 times higher than that in the control group and PMI hydrogel-treated group, respectively” to “It was found that the relative Na¹⁸F accumulation in the PTK hydrogel-treated group was higher than that in the control group and PMI hydrogel-treated group” for Fig 5d.*

REVIEWERS' COMMENTS

Reviewer #4 (Remarks to the Author):

Having thoughtfully considered the authors' response, I am still struggling to see how the PET study provides any evidence of the efficacy of the cartilage regeneration treatment. Having looked a second time at the author-provided data (see attached slide #1), I could not find any visible differences between the two knees within the same animal or between animals. In general, region-of-interest analysis is only useful to quantify differences that can be visualized. The method is sensitive to ROI placement and, in the absence of an objective and unbiased approach to place ROIs, I would be wary of ROI findings that are not supported by visual inspection of the images.

There are several reasons why the PET study could be inconclusive. First the cartilage defects are very small (2x2 mm), especially compared to the resolution of the scanner which is likely around 1.5 mm. Second, the normal surrounding bone has very high uptake, which overshadows the signal from within the cartilage defect. Third, in the absence of co-registered CT images, the exact location of the cartilage defect is almost impossible to locate on the PET images in an objective and unbiased manner.

The values reported in the bar plot in Fig 5d,e cannot be justified on the basis of the raw PET data, and I would strongly advise against including these ROI data in the manuscript as they do not reflect the PET data. I am not sure how these ROI data were obtained, but my guess is that they may have been issues with ROI placement and/or slice thickness adjustments.

As to this latter point, the author continues to claim that the only change in Fig 5c between the initial submission and the subsequent revision is that they adjusted the intensity scale to better highlight the uptake in the knee ("all the changes for Fig 5c were based on the unified quantitative display settings in different groups"). This statement is demonstrably false. The images were clearly altered by narrowing the slice of interest in the maximum-intensity projection (MIP) representation of the image, such that only the injured knee is visible in the image. The healthy contralateral knee is now out of the picture, which prevents a visual comparison of the two knees. As shown in the attached slides 2 and 3, I was able to reproduce the images from the revised manuscript simply by adjusting the slice thickness. The intensity scale was unchanged.

In view of the high standards of Nature Communications, I am not able to endorse this manuscript for publication. The caveat that the authors are proposing to include ("It was found that the relative Na¹⁸F accumulation in the PTK hydrogel-treated group was higher than that in the control group and PMI hydrogel-treated group") is insufficient in my view. At the minimum, the PET study ought to be moved to the SI section with the caveat clearly noted that the results were inconclusive, or it should be removed altogether.

Reviewer #4:

Having thoughtfully considered the authors' response, I am still struggling to see how the PET study provides any evidence of the efficacy of the cartilage regeneration treatment. Having looked a second time at the author-provided data (see attached slide #1), I could not find any visible differences between the two knees within the same animal or between animals. In general, region-of-interest analysis is only useful to quantify differences that can be visualized. The method is sensitive to ROI placement and, in the absence of an objective and unbiased approach to place ROIs, I would be wary of ROI findings that are not supported by visual inspection of the images.

There are several reasons why the PET study could be inconclusive. First the cartilage defects are very small (2x2 mm), especially compared to the resolution of the scanner which is likely around 1.5 mm. Second, the normal surrounding bone has very high uptake, which overshadows the signal from within the cartilage defect. Third, in the absence of co-registered CT images, the exact location of the cartilage defect is almost impossible to locate on the PET images in an objective and unbiased manner.

The values reported in the bar plot in Fig 5d,e cannot be justified on the basis of the raw PET data, and I would strongly advise against including these ROI data in the manuscript as they do not reflect the PET data. I am not sure how these ROI data were obtained, but my guess is that there may have been issues with ROI placement and/or slice thickness adjustments.

As to this latter point, the author continues to claim that the only change in Fig 5c between the initial submission and the subsequent revision is that they adjusted the intensity scale to better highlight the uptake in the knee ("all the changes for Fig 5c were based on the unified quantitative display settings in different groups"). This statement is demonstrably false. The images were clearly altered by narrowing the slice of interest in the maximum-intensity projection (MIP) representation of the image, such that only the injured knee is visible in the image. The healthy contralateral knee is now out of the picture, which prevents a visual comparison of the two knees. As shown in the attached slides 2 and 3, I was able to reproduce the images from the revised manuscript simply by adjusting the slice thickness. The intensity scale was unchanged.

In view of the high standards of Nature Communications, I am not able to endorse this manuscript for publication. The caveat that the authors are proposing to include ("It was found that the relative Na¹⁸F accumulation in the PTK hydrogel-treated group was higher than that in the control group and PMI hydrogel-treated group") is insufficient in my view. At the minimum, the PET study ought to be moved to the SI section with the caveat clearly noted that the results were inconclusive, or it should be removed altogether.

Our reply: *Thanks for your comments. Based on your concerns, we have completely removed the PET results from the revised manuscript and supplementary materials, which does not change our major conclusion that PTK could achieve a full-thickness*

cartilage regeneration in vivo and may provide a new solution to address the problem of cartilage regeneration.